# HiCoLoRA: Addressing Context-Prompt Misalignment via Hierarchical Collaborative LoRA for Zero-Shot DST

## Abstract

Zero-shot Dialog State Tracking (zs-DST) is essential for enabling Task-Oriented Dialog Systems (TODs) to generalize to new domains without costly data annotation. A central challenge lies in the semantic misalignment between dynamic dialog contexts and static prompts, leading to inflexible cross-layer coordination, domain interference, and catastrophic forgetting. To tackle this, we propose Hierarchical Collaborative Low-Rank Adaptation (HiCoLoRA), a framework that enhances zero-shot slot inference through robust prompt alignment. It features a hierarchical LoRA architecture for dynamic layer-specific processing (combining lower-layer heuristic grouping and higher-layer full interaction), integrates Spectral Joint Domain-Slot Clustering to identify transferable associations (feeding an Adaptive Linear Fusion Mechanism), and employs Semantic-Enhanced SVD Initialization (SemSVD-Init) to preserve pre-trained knowledge. Experiments on multi-domain datasets MultiWOZ and SGD show that HiCoLoRA outperforms baselines, achieving SOTA in zs-DST. Code is available at Anonymous Github.

## 1 Introduction

Task-Oriented Dialog Systems (TODs) help users complete specific tasks, such as restaurant reservations or taxi inquiries, through multi-turn natural language interactions Luo et al. (2024); Wang et al. (2024d). A core component enabling this functionality is Dialog State Tracking (DST), which dynamically parses user inputs into structured slot-value pairs to infer intents and resolve ambiguities. However, zero-shot DST (zs-DST) faces a challenge: semantic misalignment between dynamic dialog contexts and static prompts, hindering adaptation to new domains.

To extend DST modules to unseen domains by leveraging existing knowledge and address data scarcity, zs-DST has emerged as a promising paradigm. While approaches include data augmentation He et al. (2025) and prompt engineering Liu et al. (2025b); Wang et al. (2024c); Aksu et al. (2023), parameter-efficient fine-tuning (PEFT), particularly Low-Rank Adaptation (LoRA) Wang et al. (2024a); Occhipinti et al. (2024), has gained prominence for zs-DST Yi et al. (2025); Aksu et al. (2023). LoRA freezes most pre-trained model parameters, updating only low-rank external matrices to enable efficient cross-domain generalization. Recent multi-LoRA variants, such as DualLoRA Luo et al. (2024), CoLA Zhou et al. (2025), HydraLoRA Tian et al. (2024), MTL-LoRA Yang et al. (2025), enhance adaptability through specialized adapters or cross-task collaboration. Despite these advances, structural limitations in context-prompt alignment persist, motivating our hierarchical approach. Despite these advancements, current LoRA based zs-DST methods face limitations. Data augmentation and prompt engineering approaches manipulate external data or rely on shallow input adjustments, which may not adequately cover a broad range of slot types or capture the nuanced complexities of dynamic dialog contexts. Similarly, PEFT methods rely on shallow input adjustments or local parameter modifications, which limits their adaptability to complex and dynamic dialog contexts. Specifically, a single LoRA project features different tasks in the same low-dimensional space. This can lead to intertask interference, hinder knowledge separation, and limit multitask adaptability. Although Multi-LoRA designs like DualLoRA Luo et al. (2024), CoLA Zhou et al. (2025), HydraLoRA Tian et al. (2024), and MTL-LoRA Yang et al. (2025) have attempted to mitigate these issues by introducing multipath adapters or exploring adaptive cross-task collaboration, limitations persist in practical applications. These limitations stem largely from a

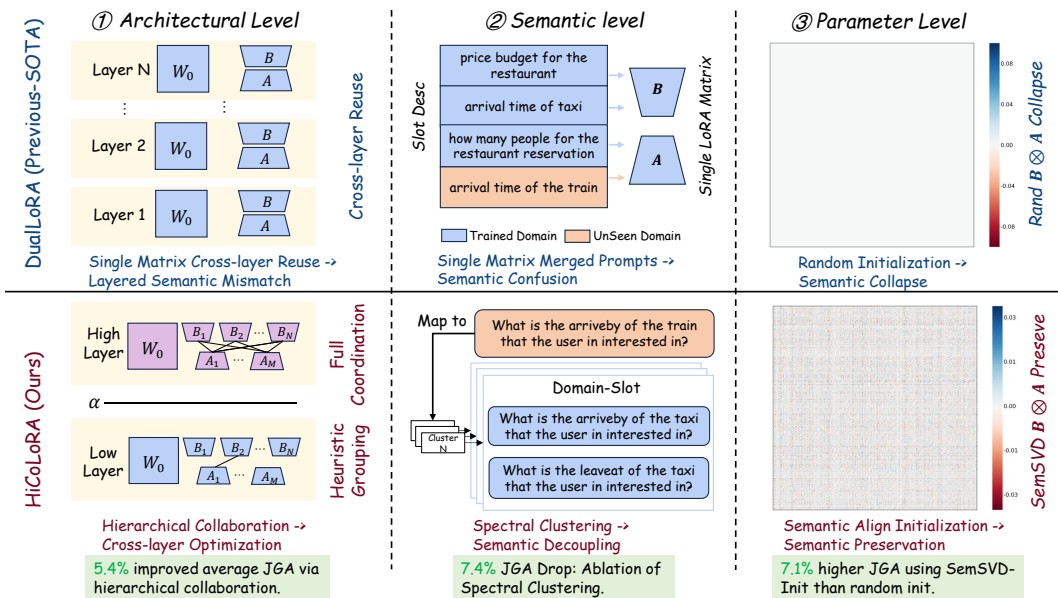

Figure 1: Three critical challenges motivating our work: (1) Architectural rigidity hinders cross-layer coordination in Transformers, limiting fine-grained semantic alignment; (2) Coupling of domain-shared and domain-specific semantics causes cross-domain confusion; (3) Random parameter initialization distorts pre-trained knowledge, exacerbating catastrophic forgetting.

structural mismatch between dynamic dialog contexts and static prompts (as illustrated in Fig.1), manifesting itself in three critical research challenges: (RQ1) Rigid hierarchical designs hinder effective cross-layer weight sharing, limiting fine-grained semantic alignment in deeper layers. (RQ2) A single adaptation matrix conflates domain-agnostic and domain-specific signals, causing semantic confusion between domains. (RQ3) The use of random initialization for LoRA parameters can distort pre-trained knowledge and exacerbate catastrophic forgetting.

To address the three limitations, we propose Hierarchical Collaborative Low-Rank Adaptation (HiCoLoRA), a novel framework inspired by DualLoRA's prompt augmentation Luo et al. (2024) and CoLA's multi-LoRA grouping Zhou et al. (2025). Departing from "uniform layer processing", it introduces: (1) A Hierarchical Collaborative Architecture with lower-layer heuristic grouping and higher-layer full interaction, resolving RQ1 via dynamic cross-layer coordination; (2) Spectral Joint Clustering and Adaptive Fusion disentangling domain-shared and specific semantics addressing RQ2; (3) Semantic-Enhanced SVD Initialization preserving pre-trained knowledge against RQ3.

Experiments on MultiWOZ and SGD datasets establish new SOTA results, achieving 5.4% and 9.4% JGA gains over DualLoRA, validating our framework's success in fundamentally addressing RQ1-3 through hierarchical adaptation, semantic disentanglement and knowledge preservation, especially in high-overlap domains and sparse slots.

## 2 RELATED WORK

**zs-DST and Goal Accuracy.** zs-DST is fundamental to TODs, with Joint Goal Accuracy (JGA) and Average Goal Accuracy (AGA) as a key metric to evaluate performance. Early methods like TRADE Wu et al. (2019) and SUMBT Lee et al. (2019) laid the foundations for cross-domain generalization but relied on task-specific architectures. With pre-trained language models (PLMs), SimpleTOD Hosseini-Asl et al. (2020) reformulated DST as sequence generation, Yi et al. (2025) enhanced the few-shot capability through the enhancement of intent-driven dialog information. Recent advances in zs-DST include Prompter Aksu et al. (2023) with learnable prompts, DualLoRA Luo et al. (2024) with dual-path adapters and LUAS Wang et al. (2024d) with synthetic data, though these still face challenges in cross-layer semantic alignment and domain knowledge separation. HiCoLoRA fun-

damentally optimizes these challenges through its hierarchical cross-layer coordination and spectral domain-slot disentanglement,

**Parameter-Efficient Fine-Tuning with LoRA.** LoRA is an effective method for PEFT Zhang et al. (2025); Liu et al. (2025a); Jabbarvaziri & Lampe (2025); Zhang & Pilanci (2025); Wang et al. (2024b). For zero-shot scenarios, DualLoRA Luo et al. (2024) mitigates context-prompt misalignment via dual-path designs, while HydraLoRA Tian et al. (2024) uses MoE routers for subtask decoupling. Multitask adaptations such as CoLA Zhou et al. (2025) and MTL-LoRA Yang et al. (2025) enhance cross-task collaboration, and RoSA Nikdan et al. (2024) integrate routing/sparsity for efficiency. Initialization strategies such as PiSSA Meng et al. (2024), MiLoRA Zhang et al. (2024), improve knowledge preservation, but few address semantic alignment in dynamic dialog contexts. HiCoLoRA directly addresses this gap through hierarchical cross-layer semantic coordination and adaptive domain-slot disentanglement, enabling alignment in dynamic dialog contexts.

**Layer-Specific Algorithms in Transformers.** Xie et al. (2025); Wang et al. (2025); Liu et al. (2024); Du et al. (2020) all demonstrate that the lower layers handle basic and detailed information, such as lexical semantics, grid features, and rapid computations, while the upper layers focus on abstract and task-oriented processing, such as prediction, abstract planning, and semantic integration. Algorithm designs targeting this characteristic have improved task performance. Layer-specific algorithms also include Split Attention (partitioning attention across layers) Lin et al. (2025), Hierarchical LoRA (applying hierarchical LoRA patterns) Xiao et al. (2024); Guo et al. (2024), Dynamic Layer Replace (selective layer substitution) Xiong et al. (2024) and attention head pruning within layers He & Lin (2025); Zayed et al. (2024). These methods leverage the principle of unequal layer contributions across tasks, achieving computational or parameter reductions while improving metrics. Critically, they fail to resolve issues such as dynamic context-prompt misalignment induced by layer-specific adaptations and the lack of coordinated cross-layer interactions needed for semantic coherence, which challenges complex scenarios in zs-DST. HiCoLoRA thus aims to bridge this gap by introducing mechanisms for harmonizing hierarchical layer-wise adaptations and ensuring consistent cross-layer alignment essential for robust zs-DST performance.

## 3 METHOD

We propose Hierarchical Collaborative Low-Rank Adaptation (HiCoLoRA, Fig. 2), a method that enhances zero-shot slot inference in unseen domains through improved prompt alignment. HiCoLoRA employs a hierarchical architecture that moves beyond uniform layer-wise processing, dynamically integrating domain-agnostic (UniRep-LoRA) and domain-specific (SemAdapt-LoRA) semantics via adaptive fusion. Additionally, spectral clustering and SemSVD-Init optimize domain-slot representations to strengthen zero-shot generalization.

### 3.1 UNIVERSAL REPRESENTATION LORA (UNIREP-LORA)

UniRep-LoRA is designed to efficiently capture domain-agnostic semantic information from the dialog context $x_{ur}$, such as universal slots for time and location. By freezing the parameters of the pre-trained model $W_0$ and updating only the low-rank matrices $B_{ur}$ and $A_{ur}$:

$$h_{ur} = W_0 x_{ur} + B_{ur} A_{ur} x_{ur}. \tag{1}$$

UniRep-LoRA and SemAdapt-LoRA are combined via adaptive linear fusion, balancing general and domain-specific representations to mitigate context-prompt misalignment for zero-shot scenarios.

### 3.2 SEMANTIC ADAPTATION LORA (SEMADAPT-LORA)

In contrast to UniRep-LoRA, which focuses on universal features, SemAdapt-LoRA is specifically tailored to optimize domain-specific prompts by dynamically adjusting their influence across different domains (relating to RQ2). To enable this domain-specific optimization, we introduce a Multi-Head Attention module: high-frequency dialog words from the train dataset serve as $Q$, while slot descriptions function as $K$ and $V$, with the output denoted as $x_{sa}$. Different attention heads allow for the capture of diverse semantic correlations between these two components. By improving the semantic alignment between high-frequency dialog information and slot descriptions, this setup

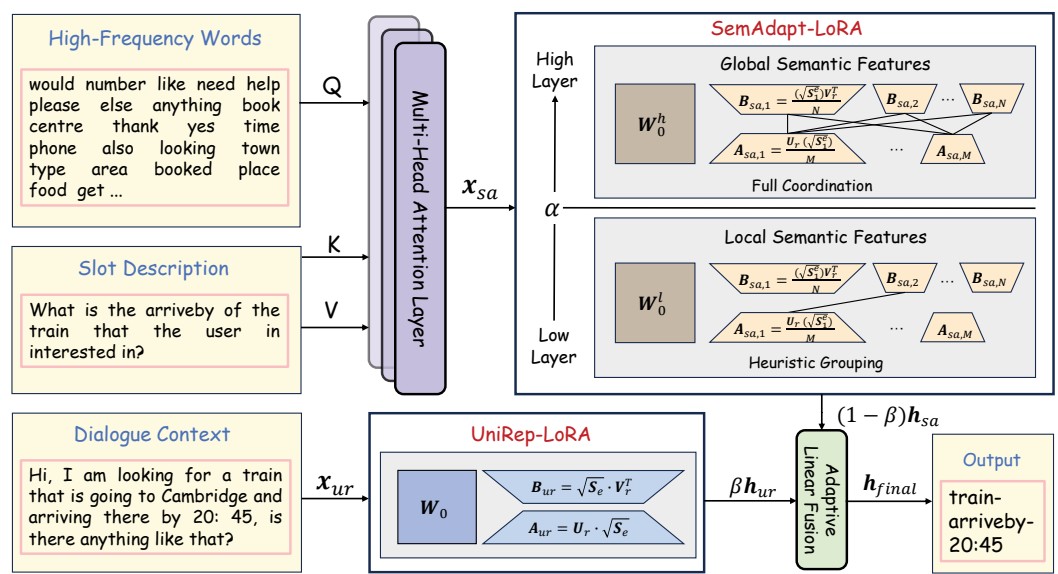

Figure 2: Overview of the HiCoLoRA framework, which combines: (1) UniRep-LoRA and SemAdapt-LoRA with Adaptive Linear Fusion balancing domain-agnostic and domain-specific features; (2) Spectral Joint Domain-Slot Clustering disentangling domain semantics to guide fusion; (3) SemSVD-Init preserving pre-trained knowledge via singular value modulation. These synergistically address context-prompt misalignment, enhancing zero-shot slot inference.

provides SemAdapt-LoRA with more effective local semantic features as input, thus supporting its hierarchical collaborative mechanism to improve zero-shot slot inference performance. To achieve such fine-grained adaptive prompt processing, we further introduce two sets of trainable matrices: $A_{sa}^m|_{m=1}^M$ for domain common prompt encoding and $B_{sa}^n|_{n=1}^N$ for cluster-specific domain-slot reconstruction. The former compresses high-dimensional prompt semantics into a low-rank space, while the latter reconstructs low-rank features based on specific domain semantics, transforming general features into domain-specific representations. In detail, $M$ is the number of clusters of domains in domain clusters and $N$ is the number of clusters of domains.

To ensure effective collaboration between $A_{sa}^m|_{m=1}^M$ and $B_{sa}^n|_{n=1}^N$, we propose a novel cross-layer collaborative module. Tailored to RQ1, it employs two interaction strategies across Transformer layers, aligned with their semantic roles: Lower layers encode Local Semantic Features serving as semantic atoms for higher layers, while higher layers model Global Semantic Features and guide Lower layer feature extraction via attention suppressing irrelevant associations. This paradigm transcends traditional Transformers' uniform layer processing, forming a hierarchical semantic chain from local associations to global intent modeling.

**Heuristic Grouping.** For lower Transformer layers, tasked with encoding local semantic atoms, heuristic grouping is favored for its efficiency. It clusters semantically similar parameters to avoid irrelevant interactions, aggregating coherent local features that serve as building blocks for higher layers, laying a precise foundation for global processing.

$$h_{sa} = W_0^l x_{sa} + N B_{sa}^* M A_{sa}^* x_{sa}, \qquad (2)$$

where $W_0^l$ denotes lower-layer weights. The optimal algorithm for selecting matrices $A_{sa}^*$ and $B_{sa}^*$ is based on calculating the cosine similarity between the average vector of the slot clusters and the slot prompts. Specifically, $A_{sa}^*$ refers to the matrix $A$ corresponding to the category with the highest similarity between the embedding groups $x_{sa}$ and the domain clusters $\mathcal{D}^M$, while $B_{sa}^*$ denotes the matrix $B$ associated with the category showing the highest similarity between $x_{sa}$ and the slot prompt clusters $\mathcal{X}^N$. During training, differentiable selection is achieved via Gumbel-Softmax based on cluster similarity, while softmax is used to accelerate during inference.

**Full Collaboration.** Higher layers model global semantic connections through full collaboration, enabling comprehensive interactions between all encoded local atoms to capture implicit associations such as the link between *train-arriveby* and *destination*. This process, combined with attention-guided noise suppression from lower layers, resolves fine-grained alignment and enhances zero-shot generalization. The equation is shown below:

$$\boldsymbol{h}_{sa} = \boldsymbol{W}_0^h \boldsymbol{x}_{sa} + \sum_{n=1}^{N} \boldsymbol{B}_{sa}^n \sum_{m=1}^{M} \boldsymbol{A}_{sa}^m \boldsymbol{x}_{sa}, \tag{3}$$

where $\boldsymbol{W}_0^h$ denotes high-layer weights.

### 3.3 Inference Efficiency Analysis

A legitimate concern with multi-branch LoRA designs is potential inference latency. HiCoLoRA addresses this through a precomputation and merging strategy that ensures inference efficiency comparable to standard LoRA.

**UniRep-LoRA Simplicity:** The UniRep-LoRA module (Eq. 1) maintains a single set of $\boldsymbol{A}_{ur}$ and $\boldsymbol{B}_{ur}$ matrices throughout.

**Heuristic Grouping:** For lower layers employing heuristic grouping (Eq. 2), only a single optimal pair $\boldsymbol{A}_{sa}^*$ and $\boldsymbol{B}_{sa}^*$ is selected, requiring just one matrix multiplication per forward pass.

**Full Collaboration:** During training, SemAdapt-LoRA employs multiple $\boldsymbol{A}_{sa}^m$ and $\boldsymbol{B}_{sa}^n$ matrices to enable fine-grained semantic adaptation. During inference, we precompute the collective low-rank contribution of all matrix pairs, and the SemAdapt-LoRA output in full collaboration layers (Eq. 3) can be reorganized by computing aggregated matrices:

$$\boldsymbol{A}_{\text{total}} = \sum_{m=1}^{M} \boldsymbol{A}_{sa}^m, \quad \boldsymbol{B}_{\text{total}} = \sum_{n=1}^{N} \boldsymbol{B}_{sa}^n \tag{4}$$

yielding the equivalent computation:

$$\boldsymbol{h}_{sa} = \boldsymbol{W}_0^h \boldsymbol{x}_{sa} + \boldsymbol{B}_{\text{total}} \boldsymbol{A}_{\text{total}} \boldsymbol{x}_{sa} \tag{5}$$

This transformation reduces the computational overhead from $O(M \cdot N)$ matrix multiplications to merely two matrix multiplications, identical to standard LoRA.

The matrix additions involved in precomputation $O(r \cdot d)$ are negligible compared to matrix multiplications $O(d^2)$. In practice, we precompute all low-rank update terms during model export and absorb them into the base model weights, eliminating any additional inference overhead. Consequently, despite its hierarchical architecture, HiCoLoRA maintains inference latency on par with standard LoRA implementations.

### 3.4 Adaptive Linear Fusion Mechanism

We introduce an adaptive linear fusion mechanism to merge the two LoRA modules. A learnable gating coefficient $\beta$, trained end-to-end, balances general and semantically adaptive features. This allows flexible integration of multi-level semantics based on dialog contexts and domain-slot prompts, more effectively resolving dynamic-static prompt mismatches than fixed-coefficient weighting.

$$\boldsymbol{h}_{final} = \beta \boldsymbol{h}_{ur} + (1 - \beta) \boldsymbol{h}_{sa}, \quad \beta \in (0, 1). \tag{6}$$

### 3.5 Spectral Clustering of Domains and Slot Prompts

The Spectral Joint Domain and Slot Clustering mechanism identifies semantic relatedness by leveraging commonalities across domains and slot prompts. Domains often exhibit categorical abstraction, such as *train* and *taxi* belonging to transportation, or *hotel* and *restaurant* representing service-oriented establishments. Slot prompts are formatted as structured {*domain-slot: question*} pairs, for instance {*train-arriveby: what is the arrival time of the train the user is interested in?*}, which helps to uncover semantic commonalities among slots from different domains. Prompts like *train-arriveby* and *taxi-arriveby* both express temporal attributes despite originating in distinct domains.

The T5 encoder converts these domain names and extended slot prompts into dense vector representations, followed by spectral clustering via Laplacian matrix eigendecomposition. The optimal number of clusters ($M$ for domains, $N$ for slot prompts) is determined by maximizing the silhouette coefficient, producing clusters $\mathcal{D}^M$ and $\mathcal{X}^N$.

### 3.6 Semantic-Enhanced SVD Initialization

Semantic-Enhanced Singular Value Decomposition Initialization (SemSVD-Init) is an approach to parameter initialization for both UniRep-LoRA and SemAdapt-LoRA modules. Unlike Kaiming LoRA initialization, which can disrupt pre-trained semantic structures, or methods like PiSSA Meng et al. (2024) that lack explicit task-specific alignment (related to RQ3), SemSVD-Init directly addresses preserving pre-trained knowledge while enhancing domain and slot related semantics, priming the model for effective zero-shot transfer.

SemSVD-Init aligns singular values with the clustered semantic space, and singular directions associated with universal semantics are amplified while those that capture domain-specific noise are suppressed. Taking the UniRep-LoRA module for example, the initialization process begins by performing SVD on the model weight matrix $\boldsymbol{W}_0$:

$$\boldsymbol{W}_0 = \boldsymbol{U}_r \boldsymbol{\Sigma}_r \boldsymbol{V}_r^T. \tag{7}$$

Subsequently, a correlation matrix $\boldsymbol{R}$ is computed by cosine similarity between the right singular vectors $\boldsymbol{V}_r$ and the cluster embeddings $\text{T5}_{\text{en}}(\mathcal{X}^N)$, where $\text{T5}_{\text{en}}$ denotes the embeddings of the encoder of the T5 model.

$$\boldsymbol{R} = \cos(\boldsymbol{V}_r, \text{T5}_{en}(\mathcal{X}^N)). \tag{8}$$

Using these correlations, the singular values are enhanced on the basis of maximum category relevance for each vector.

$$\boldsymbol{S}_e = \text{diag}\big(\sigma_1 \cdot \text{ReLU}(1 + \lambda \boldsymbol{R}_1), \ldots, \sigma_r \cdot \text{ReLU}(1 + \lambda \boldsymbol{R}_r)\big), \tag{9}$$

where $\boldsymbol{R}_k$ is the relevance score for the $k$-th singular vector, derived from the correlation between $\boldsymbol{V}_r[:, k]$ and the cluster embeddings, $\text{ReLU}(x) = \max(0, x)$ to ensure positivity, and $\lambda$ is a hyperparameter. The LoRA matrices are initialized as:

$$\begin{aligned}
\boldsymbol{A}_{ur} &= \sqrt{\boldsymbol{S}_e} \boldsymbol{V}_r^T, \\
\boldsymbol{B}_{ur} &= \boldsymbol{U}_r \sqrt{\boldsymbol{S}_e}.
\end{aligned} \tag{10}$$

Finally, the residual weight matrix $\boldsymbol{W}_{res}$ is adjusted to preserve key knowledge of the pre-trained model and avoiding distortion of its semantic structure.

$$\boldsymbol{W}_{res} = \boldsymbol{W}_0 - \boldsymbol{B}_{ur} \boldsymbol{A}_{ur}. \tag{11}$$

## 4 Experiments

### 4.1 Experimental Setting

**Dataset.** We conducted experiments on two of the most prominent multi-domain TOD benchmark datasets (details in Appendix B.1). The MultiWOZ 2.1 dataset is a richly annotated corpus comprising more than 10,000 human-human written dialogs spanning multiple domains and topics. The Schema Guided dialog (SGD) dataset contains more than 20,000 dialogs covering 26 services across more than 20 domains. The data splitting strategy strictly segregates training and test domains.

**Baseline.** To evaluate the generalizability of the proposed HiCoLoRA method, we conduct a comparison against representative baselines and SOTA approaches (details in Appendix B.2, categorized into three groups: **Traditional Methods** including TRADE Wu et al. (2019), SGD-baseline Rastogi et al. (2019), MA-DST Kumar et al. (2020) and Seq2Seq-DU Feng et al. (2021); **Pre-trained Model Fine-Tuning Approaches** such as SUMBT Lee et al. (2019), GPT2-DST Li et al. (2021a), TransferQA Li et al. (2021b), T5DST Lin et al. (2021) and SlotDM-DST Wang et al. (2022); and **Previous Zero-Shot SOTA Methods** comprising Prompter Aksu et al. (2023), DCC Wang et al. (2023) and

| Method | Year | Base Model | Attraction | Hotel | Restaurant | Train | Taxi | Average |
|---|---|---|---|---|---|---|---|---|
| TRADE | 2019 | customized seq2seq | 20.1 | 14.2 | 12.6 | 22.4 | 59.2 | 25.7 |
| MA-DST | 2020 | TRADE | 22.5 | 16.3 | 13.6 | 22.8 | 59.3 | 26.9 |
| SUMBT | 2019 | BERT-base | 22.6 | 19.1 | 16.5 | 22.5 | 59.5 | 28.0 |
| GPT2-DST | 2021 | GPT2-base | 23.7 | 18.5 | 21.1 | 24.3 | 59.1 | 29.3 |
| T5DST | 2021 | T5-small | 31.9 | 20.7 | 20.1 | 28.8 | 64.1 | 33.1 |
| SlotDM-DST | 2022 | T5-small | 33.9 | 18.9 | 20.8 | 37.0 | 66.3 | 35.4 |
| T5DST* | 2021 | PPTOD-small | 35.5 | 20.0 | 25.3 | 35.3 | 65.6 | 36.4 |
| Prompter | 2023 | PPTOD-small | 35.8 | 19.2 | 26.0 | 39.0 | 66.3 | 37.2 |
| DCC | 2023 | T5-small | 35.8 | **24.8** | 22.9 | 40.2 | 65.9 | 37.9 |
| DualLoRA (Prev. SOTA) | 2024 | PPTOD-small | 37.1 | 18.9 | 27.9 | 42.4 | 67.2 | 38.7 |
| **HiCoLoRA (Ours)** | 2025 | PPTOD-small | **38.9** | 20.4 | **31.0** | **44.9** | **68.6** | **40.8** |
| **% Gain vs DualLoRA** | | | **+4.9** | **+7.9** | **+11.1** | **+5.9** | **+2.1** | **+5.4** |

Table 1: Zero - shot JGA (%) on the MultiWOZ dataset with relative improvement over previous SOTA. All results of baselines were reported from original papers. T5DST* was excerpted from Prompter Aksu et al. (2023).

DualLoRA Luo et al. (2024). Additionally, comparisons with recent advanced LoRA variants and larger-scale LLMs are included to thoroughly assess scalability and generalization.

**Metrics.** We evaluate all models using Joint Goal Accuracy (JGA) and Average Goal Accuracy (AGA). JGA measures the rate of turns with all slots exactly matched, indicating system-level reliability. AGA calculates the ratio of correctly predicted to total slots, accounting for missed true slots and errors, reflecting fine-grained slot recall and local semantic alignment. The metrics' formulas and additional experimental details are provided in Appendices B.3 and B.4.

## 4.2 MAIN REULTS

Performance comparisons on MultiWOZ and SGD benchmarks are presented in Table 1 and 2 (Table 2 in Appendix A.1). We have the following observations:

**Overall Performance Superiority.** HiCoLoRA achieves new state-of-the-art results on both Multi-WOZ and SGD benchmarks, with an average JGA of 40.8 on MultiWOZ and significant gains across all SGD domains. This consistent improvement is attributed to architectural advances that address key limitations of previous approaches: (1) traditional methods rely on rigid feature engineering; (2) full fine-tuning suffers from catastrophic forgetting; (3) prior SOTA models are limited by shallow prompting or uniform layer adaptation. HiCoLoRA overcomes these issues via integrated hierarchical adaptation. Furthermore, the model achieves an AGA of 93.8% on SGD Trains, underscoring its ability to preserve rare-slot knowledge through SemSVD-Init and maintain semantic specificity across layers.

**Component-Wise Efficacy Validation.** HiCoLoRA demonstrates strong performance across diverse domain types, attributed to its custom architectural components. In **transfer-rich domains** such as *Media*, the model achieves a JGA of 75.9%, representing a 9.4% improvement over DualLoRA. This gain is facilitated by spectral clustering, which effectively identifies cross-domain semantic commonalities, exemplified by shared attributes such as genre, thereby disentangling domain-shared semantics and mitigating signal conflation. In **domain-specific regimes** such as *Hotel*, HiCoLoRA attains JGA 20.4%, corresponding to a 7.9% relative improvement. This enhancement stems from the semantic-enhanced singular value modulation within SemSVD-Init, which preserves sparse slot semantics that are otherwise distorted under random initialization. For **context-sensitive domains** like *Messaging*, where performance is inherently limited by slot boundary ambiguities, the adaptive fusion mechanism dynamically balances static prompts against volatile dialog contexts, yielding a 4.0% gain over the rigid weighting strategy employed by DualLoRA.

**Architectural Validation Against Prev. SOTA.** The hierarchical design of HiCoLoRA directly addresses core limitations of DualLoRA. **Cross-Layer Rigidity (RQ1)**: DualLoRA's uniform processing hinders fine-grained alignment. HiCoLoRA's heuristic grouping (lower layers) and full collaboration (higher layers) enable dynamic coordination, boosting *Restaurant* JGA to +11.1%. **Semantic Conflation (RQ2)**: Where DualLoRA's single adaptation matrix confuses domain signals,

spectral joint clustering separates transport domain semantics (*Taxi*: 44.9 JGA, +2.1% error reduction). **Knowledge Distortion (RQ3)**: DualLoRA's random initialization loses rare slot knowledge. SemSVD-Init preserves pre-trained semantics, critical for *Flights'* technical terms JGA +8.1%.

**Discussion.** HiCoLoRA fundamentally resolves context prompt misalignment via hierarchical adaptation, spectral semantic disentanglement, and knowledge preserving initialization. By overcoming DualLoRA's structural limitations, our method establishes a new paradigm for zs-DST. Future work will address extreme sparse slots through domain aware initialization refinements.

## 4.3 ABLATION STUDY

We conduct an ablation study (Table 6 in Appendex A.5) to assess the contribution of each key component of HiCoLoRA.

**w/o Swap Hierarchical Strategies** swap layer-wise strategies, using heuristic grouping in high layers and full collaboration in low layers. This variant sees an 8.3% drop in the average JGA. The decline arises because it disrupts synergy: lower layers are designed to capture local semantic atoms, while higher layers model global intents. Swapping strategies break this division, validating the assumption that layer-specific roles are critical for performance.

**w/o Adaptive Linear Fusion** replaces adaptive gating with DualLoRA's static $\beta = 0.5$, causing a 12.0% JGA drop, notably in Attraction and Train domains. This exacerbates that static weighting cannot dynamically balance UniRep-LoRA (domain-agnostic) and SemAdapt-LoRA (domain-specific) features across layers. Unlike the adaptive mechanism that mitigates cross-layer semantic mismatches, static $\beta$ locks in misalignment, leading to performance drops.

**w/o Spectral Joint Cluster** discards spectral clustering, retaining the same number of $M$ and $N$ but without identifying transferable domain-slot associations. Its average JGA drops 7.4%, notably in Train and Taxi domains. The decline occurs because spectral clustering captures cross-domain semantic commonalities, such as "arriveby" in trains and taxis sharing temporal attributes, to guide effective feature fusion. Without it, the model fails to leverage transferable associations, weakening the alignment between domain-slot prompts and dynamic contexts, thus hindering zero-shot generalization.

**w/ Kaiming Init** use Kaiming initialization for matrix $A$ and zero initialization for matrix $B$ results 6.6% decreased the average JGA. SemSVD-Init preserves pre-trained semantics by modulating singular values, thereby suppressing catastrophic forgetting. Without this mechanism, random initialization induces knowledge distortion and forgetting, preventing the model from retaining critical semantics and impairing its zero-shot transfer capability.

**w/ PiSSA Init** use PiSSA initialization, trailing HiCoLoRA by 4.7% but outperforming random init. PiSSA partially addresses RQ3 but not as effectively: it retains pre-trained knowledge but lacks alignment of singular values to domain-slot semantics, limiting performance.

**w/ MiLoRA Init** use MiLoRA initialization, resulting in a significant performance drop. This degradation occurs because the MiLoRA strategy, which is designed to update minor singular components, is misaligned with the limited parameter capacity and the flat singular value spectrum of the T5-small model. Consequently, it fails to preserve crucial pre-trained semantics and severely impairs the model's zero-shot transfer capability.

Ablation studies demonstrate that the hierarchical collaborative architecture, adaptive fusion, spectral clustering, and SemSVD-Init components of HiCoLoRA are all indispensable. These components synergistically address the three core research questions, outperform baselines in zs-DST, and thus validate the efficacy of the proposed design.

## 5 ANALYSIS

This section evaluates HiCoLoRA design choices to validate its mechanisms, including rank sensitivity, high layer ratio, and attention alignment (Figs. 3–5), examining expressiveness balance, semantic flow optimization, and sustained attention for zero-shot performance.

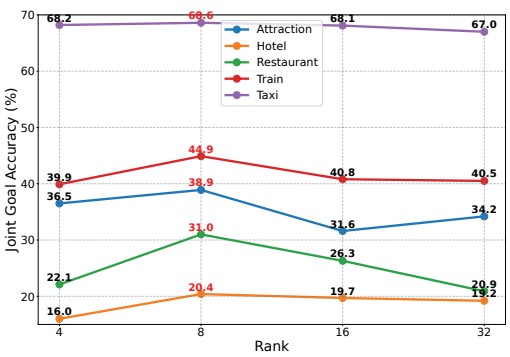

Figure 3: Accuracy of HiCoLoRA with different rank on the MultiWOZ dataset.

Figure 4: Accuracy of different high layer ratio (full collaboration) in HiCoLoRA.

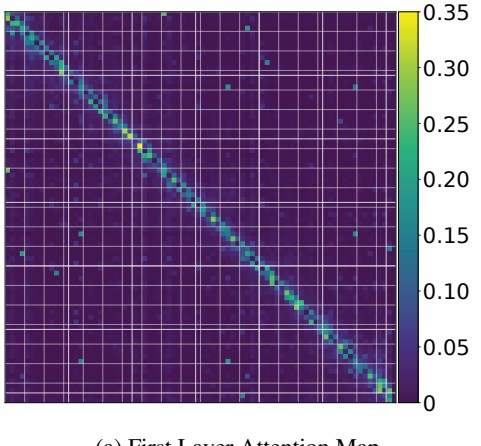

(a) First Layer Attention Map

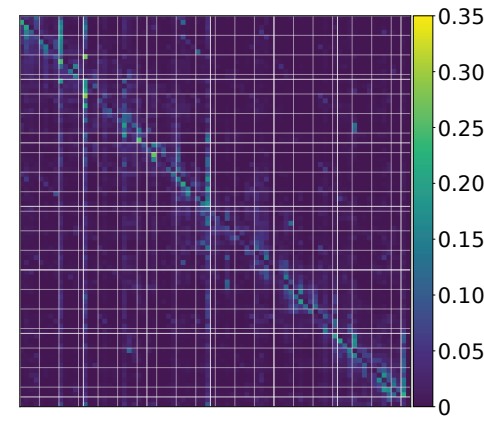

(b) Last Layer Attention Map

Figure 5: Example Attention Maps of the First and Last Transformer Layers in HiCoLoRA.

### 5.1 MODEL MECHANISM ANALYSIS

**Rank Sensitivity: Balance of expressiveness.** Fig. 3 shows that the superiority of $rank = 8$ reflects LoRA principles: the rank must match the semantic complexity. Too low (4) fails to encode nuanced domain slot distinctions. Too high (16/32) introduces redundancy and dilutes transferable signals. This aligns with low-rank matrix theory, where rank determines perturbation precision to pre-trained weights, optimizing zero-shot transfer by balancing parsimony and expressiveness.

**High-Layer Ratio: Optimizing Semantic Flow.** Fig. 4 indicates that the 50% high-layer ratio validates cognitive theories of dialog comprehension, requiring balanced local-global integration. The 0% ratio ignores global intent; 100% dilutes slot-specific cues. HiCoLoRA's hierarchical design mirrors bottom-up (local atoms) to top-down (global intent) processing, ensuring coherent semantic chains, critical to resolving dynamic context-prompt misalignment in zs-DST.

**Attention Alignment: Maintaining Semantic Focus.** Fig. 5 reveals hierarchical attention evolution: first-layer "local dots" encode discrete context-prompt associations, while last-layer "connected lines" form global semantic chains. This mirrors the layered semantic progression of Transformer: lower layers anchor atomic prompt-semantic links, and higher layers integrate into coherent intent pathways through cross layer optimization. By preserving prompt focus across depths, HiCoLoRA avoids deep-layer attention dilution, maintaining critical alignment for zero-shot transfer.

The experimental results here validate our claims: optimal rank 8 confirms balanced expressiveness, the 50% high-layer ratio verifies the optimization of semantic flow, and attention evolution demonstrates effective hierarchical collaboration. These align with the HiCoLoRA design, proving that its components jointly resolve misalignment.

## 5.2 CASE STUDY SUMMARY

Our case study analysis in Appendix C reveals HiCoLoRA's strengths in handling complex multi-domain dialogues through hierarchical collaboration and semantic disentanglement. Successful cases demonstrate robust slot inference in both transfer rich and context sensitive domains. However, failure patterns highlight areas for future refinement, particularly in highly idiosyncratic domains.

## 5.3 EXTENDED EXPERIMENTAL ANALYSIS

**Scalability Analysis.** HiCoLoRA exhibits enhanced scalability in larger datasets: 9.4% average JGA gain in SGD vs 5.4% in MultiWOZ. This stems from: 1) Semantic regular domains like *Media* benefit from spectral clustering's cross- ervice pattern recognition; 2) Terminology intensive domains such as *Flights* leverage SemSVD-Init's knowledge preservation; 3) Sparsely distributed slots like *hotel-star* benefit from hierarchical refinement and singular value modulation.

**Extended Comparative Analysis.** We conduct extensive comparisons against both recent LoRA methods and largeer LLMs based approaches. As detailed in Appendix A.2 to A.4, HiCoLoRA consistently outperforms recent LoRA variants in nearly all domains, achieving the highest average JGA. This superiority underscores the effectiveness of our hierarchical adaptation and semantic aware initialization in mitigating cross layer misalignment and knowledge distortion. Furthermore, when scaled to larger backbone models, HiCoLoRA remains highly competitive with other LLM-based zs-DST methods, even surpassing the previous SOTA FnCTOD, demonstrating its generalizability across model scales. These results confirm HiCoLoRA offers a robust and scalable solution for zero-shot dialog state tracking, effectively balancing performance and parameter efficiency.

**Architectural Implications beyond Homogeneous Baselines** The comparative analysis with heterogeneous methods reveals distinctive advantages of HiCoLoRA's design philosophy. While LDST relies on full fine-tuning of LLMs and CAPID introduces additional complexity through separate prompt generation, our approach demonstrates that a unified hierarchical architecture with collaborative adapters suffices to achieve competitive performance. This underscores the significance of structural alignment with task hierarchies over merely scaling model capacity or pipeline complexity, positioning HiCoLoRA as a resource efficient yet powerful paradigm for dialog state tracking.

**Generalization Analysis.** As analysis in Appendix A.7, our model demonstrates generalization capabilities in cross domain adaptation and long tailed recognition scenarios. It achieves performance improvements on multiple datasets, underscoring its ability to transfer knowledge across diverse domains. In addition, it exhibits remarkable robustness in tail classes, effectively mitigating the performance disparity between head and tail categories. This is attributed to our framework's ability to learn a more balanced and generalizable feature representation, which prevents overfitting to the dominant head classes and fosters a more robust decision boundary for underrepresented tail classes, thereby enhancing overall model generalization in real world and long tailed environments.

The extended analyses collectively affirm that HiCoLoRA's hierarchical adaptation transcends mere parameter efficiency by fundamentally restructuring semantic flow dynamics across transformer layers. Its spectral disentanglement mechanism effectively decouples domain agnostic and domai - specific semantics, enabling robust knowledge transfer even under significant distribution shifts. This architectural paradigm demonstrates that task aligned inductive biases, rather than sheer model scale or pipeline complexity, constitute the pivotal factor for achieving scalable zero-shot generalization in dynamic dialogue environments.

## 6 CONCLUSION

zs-DST is crucial for scalable TODs but remains challenged by insufficient cross-layer coordination, semantic conflation across domains, and corruption of pre-trained knowledge. HiCoLoRA overcomes these issues via a hierarchical LoRA design for dynamic context-prompt alignment, spectral clustering for domain-slot disentanglement, and SemSVD-Init for knowledge-preserving fine-tuning. Evaluations in MultiWOZ and SGD show that HiCoLoRA significantly outperforms previous SOTA approaches, improving average JGA by 5.4% and 9.4%, respectively. Limitations remain in highly idiosyncratic slot domains, and future work will focus on slot aware refinement to further strengthen HiCoLoRA's applicability in zs-DST.

ETHICS STATEMENT

Our research involves only publicly available, anonymized dialog datasets (MultiWOZ and SGD) and does not collect new human subject data. All data usage complies with the original licenses, and no personally identifiable information is processed or stored. The proposed method, HiCoLoRA, is designed to improve zero-shot generalization in task-oriented dialog systems and does not have known harmful applications. We acknowledge that there are no conflicts of interest and that the research was conducted with full integrity, transparency, and respect for privacy, fairness, and inclusivity. No institutional review board (IRB) approval was required as the study involves no human participants beyond the use of existing, de-identified benchmark data.

REPRODUCIBILITY STATEMENT

To ensure reproducibility, we provide a complete description of HiCoLoRA's architecture and hyperparameters in Sections 3 and Appendix B.4. The anonymized source codes and datasets are publicly available at `https://anonymous.4open.science/r/HiCoLoRA-96EB`. Random seeds, optimizer settings, and model checkpoints are fully specified to enable exact replication of our results.

LLM USE STATEMENT

We acknowledge the use of Writeful integrated with Overleaf for refining the textual expression of this manuscript, and DeepSeek V3.1 for error correction of the experimental code. The role of these LLMs was limited to technical assistance and did not involve research ideation or the creation of core content, thus not meeting the defined criteria of a "contributor" ". All LLM outputs have been rigorously verified by the authors, who bear full responsibility for the final accuracy, integrity, and originality of the content including the avoidance of plagiarism or scientific misconduct.

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

## A ADDITIONAL EXPERIMENTAL RESULTS

### A.1 PERFORMANCE ON SGD DATASET

Table 2 presents the zero-shot performance of HiCoLoRA on the SGD Dataset. Compared to baseline methods and previous state-of-the-art approaches, HiCoLoRA achieves significant improvements across multiple domains.

| Method | Year | Buses | Events | Flights | Media | Messaging | Music | Payment | Trains |
|---|---|---|---|---|---|---|---|---|---|
| SGD-baseline | 2019 | 9.7/50.9 | 23.5/57.9 | 23.9/65.9 | 18.0/30.8 | 10.2/20.0 | 15.5/39.9 | 11.5/34.8 | 13.6/63.5 |
| Seq2seq-DU | 2021 | 16.8/N | 31.9/N | 15.9/N | 23.1/N | 4.9/N | 12.3/N | 7.2/N | 16.8/N |
| Transfer-QA | 2021 | 15.9/63.6 | 15.6/56.8 | 3.59/42.9 | 30.2/67.5 | 13.3/37.9 | 8.9/62.4 | 24.7/60.7 | 17.4/64.9 |
| SlotDM-DST | 2022 | 43.9/86.3 | – | – | – | 36.6/61.4 | – | 16.5/62.0 | 46.7/86.9 |
| T5DST | 2021 | 46.8/N | 48.8/N | – | 55.5/N | 59.2/N | – | 23.3/N | 53.0/N |
| Prompter | 2023 | 48.4/N | 51.5/N | – | 65.3/N | 59.2/N | – | 21.9/N | 50.8/N |
| DCC | 2023 | – | – | – | – | 28.8/N | – | 19.4/N | 42.3/N |
| DualLoRA (Prev. SOTA) | 2024 | 50.9/88.8 | 46.5/82.8 | 28.4/76.9 | 69.7/88.7 | 65.1/85.5 | 32.5/72.4 | 21.2/**70.2** | 52.9/89.3 |
| HiCoLoRA (Ours) | 2025 | **54.0/93.2** | **55.1/87.8** | **30.7/82.3** | **75.9/95.8** | **67.7/88.1** | **35.8/78.9** | **26.7**/65.0 | **55.8/93.8** |
| **% Gain vs DualLoRA** | - | **+6.1/+5.0** | **+18.5/+6.0** | **+8.1/+7.0** | **+8.9/+8.0** | **+4.0/+3.0** | **+10.2/+9.0** | **+25.9/-7.4** | **+5.5/+5.0** |

Table 2: Zero-shot JGA (%) & AGA (%) on the SGD dataset with relative improvements over previous SOTA. "N" indicates unreported results.

| Method | Year | Attr. | Hotel | Rest. | Train | Taxi | AVG. |
|---|---|---|---|---|---|---|---|
| HydraLoRA | 2024 | 35.1 | 18.9 | 26.3 | 41.5 | 65.2 | 37.4 |
| LoRA-GA | 2024 | 33.8 | 19.2 | 24.7 | 42.8 | 64.1 | 36.9 |
| RoSA | 2024 | 36.5 | 19.6 | 27.9 | 43.2 | 66.8 | 38.8 |
| Spectral Adapter | 2025 | 37.2 | 20.1 | 28.5 | 43.6 | 67.3 | 39.3 |
| HiCoLoRA (Ours) | 2025 | **38.9** | **20.4** | **31.0** | **44.9** | **68.6** | **40.8** |

Table 3: Comparison of HiCoLoRA with recent LoRA-based methods on MultiWOZ (JGA %).

## A.2 COMPARISON WITH CONTEMPORARY LORA METHODS

To situate HiCoLoRA within the evolving landscape of PEFT methods, we compare it against four contemporary LoRA variants: HydraLoRA Tian et al. (2024), LoRA-GA Wang et al. (2024b), RoSA Nikdan et al. (2024), and Spectral Adapter Zhang & Pilanci (2025). As shown in Table 3, HiCoLoRA achieves the highest average JGA, outperforming all baselines in nearly all domains. This superiority is not merely incremental; it stems from fundamental architectural and semantic distinctions that address the core challenges of zs-DST.

**Structural Design Philosophy:** While HydraLoRA introduces an asymmetric LoRA structure to enhance expressiveness, and RoSA combines low-rank and sparse adaptations for robustness, both methods retain a *layer-agnostic* approach to adapter deployment. In contrast, HiCoLoRA's *hierarchical layer-specific processing* explicitly models the divergent roles of lower and higher Transformer layers, local feature encoding versus global intent integration, enabling dynamic cross-layer coordination that is critical for resolving context-prompt misalignment.

**Semantic Alignment Mechanism:** Spectral Adapter leverages spectral initialization to better preserve pre-trained knowledge, similar to our SemSVD-Init. However, it lacks HiCoLoRA's *spectral joint clustering* of domains and slots, which actively disentangles domain-shared and domain-specific semantics. This clustering guides the adaptive fusion of general and domain-aware features, a mechanism absent in other methods, leading to more precise slot inference in transfer-rich domains like *Media*.

**Knowledge Preservation and Transfer:** LoRA-GA improves the alignment of the gradient during initialization to accelerate convergence but does not explicitly modulate the singular values to align with the specific semantics of the task. HiCoLoRA's SemSVD-Init not only preserves pre-trained knowledge, but also amplifies singular components relevant to domain-slot structures, effectively mitigating catastrophic forgetting and enhancing zero-shot generalization, particularly for rare slots such as *hotel-stars*.

**Adaptability to Dynamic Contexts:** Unlike RoSA and HydraLoRA, which are designed for general NLP tasks, HiCoLoRA is tailored for the dynamic and multi-turn nature of dialog systems. Its *adaptive gating mechanism* dynamically balances domain-agnostic and domain-specific features per turn, enabling robust handling of evolving dialog contexts, a capability that static LoRA variants lack.

| Method | Year | Base Model | Attr. | Hotel | Rest. | Train | Taxi | AVG. |
|---|---|---|---|---|---|---|---|---|
| ChatGPT-zsTOD | 2023 | ChatGPT (GPT-3.5) | 52.7 | 42.0 | 60.8 | 70.9 | 55.8 | 56.4 |
| ChatGPT-zsTOD | 2023 | ChatGPT (GPT-3.5) | 67.2 | 37.6 | 67.3 | **74.4** | 60.1 | 61.3 |
| D0T | 2024 | LLAMA2-13B | 63.1 | 43.8 | 60.8 | 48.8 | 64.7 | 56.2 |
| MoPE | 2024 | ChatGLM-6B | 60.4 | 34.1 | **64.0** | 71.3 | 55.9 | 57.1 |
| FnCTOD | 2024 | ChatGPT (GPT-4) | 58.8 | 45.2 | 69.5 | 76.4 | 63.2 | **62.6** |
| FnCTOD | 2024 | LLAMA2-13B | 62.2 | **46.8** | 60.9 | 67.5 | 60.3 | 59.5 |
| Multi-User | 2025 | GPT-4o | 56.8 | 46.0 | 61.9 | 69.3 | 55.1 | 57.8 |
| HiCoLoRA | 2025 | LLAMA2-13B | 62.0 | 42.0 | 61.0 | 65.0 | 69.0 | 60.0 |
| HiCoLoRA | 2025 | Qwen2.5-14B-Instruct | **64.0** | 44.0 | 63.0 | 68.0 | **71.0** | 62.0 |

Table 4: Zero-shot JGA (%) on MultiWOZ using large language models. HiCoLoRA demonstrates strong scalability and generalization across model scales. All results of baselines were reported from original papers.

HiCoLoRA addresses the unique challenges of zs-DST: cross-layer misalignment, semantic conflation, and knowledge distortion. While other LoRA variants offer general-purpose efficiency, HiCoLoRA provides a *domain-aware* and *layer-conscious* design that is essential for robust zero-shot transfer in TODs.

### A.3 SCALABILITY ANALYSIS: GENERALIZATION ACROSS MODEL SCALES

To rigorously assess the scalability and architectural generality of HiCoLoRA, we extend our evaluation to LLM, comparing against contemporary LLM-based zs-DST methods, including ChatGPT-zsTOD Heck et al. (2023), D0T Finch & Choi (2024), MoPE Tang et al. (2024), FnCTOD Li et al. (2024) and Multi-User Song et al. (2025). As shown in Table 4, HiCoLoRA achieves competitive performance when deployed in LLAMA2-13B and Qwen2.5-14B-Instruct, with an average JGA of 62.0% in the latter, only marginally below FnCTOD with GPT-4 (62.6%) and significantly outperforms other baselines based on LLM.

**Architectural Generalization Beyond Scale.** The consistent performance of HiCoLoRA in both both small (T5-small, 60M) and large (13B–14B) models underscores a key insight: its hierarchical adaptation mechanism is *scale-agnostic*. The efficacy of HiCoLoRA stems from its structured semantic alignment decomposition, which addresses cross-layer coordination (RQ1), domain-slot disentanglement (RQ2), and knowledge preservation (RQ3) through explicit inductive biases. This allows it to be generalized effectively even when applied to larger models without architecture-specific modifications.

**Efficiency-Performance Trade-off.** While FnCTOD benefit from extreme scale and extensive pre-training as GPT-4-based methods, HiCoLoRA offers a more efficient alternative, achieving comparable performance with only partial parameter updates. This highlights its suitability for scenarios where full fine-tuning or inference with very large models is prohibitive. The fact that HiCoLoRA outperforms other PEFT-based LLM methods further validates its superior design in leveraging limited tunable parameters for maximal semantic alignment.

**Limitations and Future Directions.** The remaining gap between the HiCoLoRA and GPT-4-based methods suggests that scale still matters to capture extremely nuanced or idiosyncratic slot semantics. However, HiCoLoRA's strong performance in structured domains such as *Restaurant* indicates that its hierarchical and spectral mechanisms effectively compensate for scale limitations through better semantic organization. Future work may explore hybrid approaches that integrate HiCoLoRA's alignment mechanisms with larger foundation models for even stronger zero-shot generalization.

### A.4 EXTENDED COMPARISON WITH FNCTOD

Since FnCTOD (Li et al., 2024) achieves comparable performance to HiCoLoRA under the same LLaMA2-13B backbone, we conduct a detailed comparison to highlight their differences in exper-

| Method | Attr. | Hotel | Rest. | Train | Taxi | AVG. | Relative Change |
|---|---|---|---|---|---|---|---|
| FnCTOD (Fine-tuned LLaMA2-13B) | 62.2 | 46.8 | 60.3 | 60.9 | 67.5 | 59.5 | -0.8 |
| FnCTOD (No FT LLaMA2-13B) | 49.8 | 29.5 | 48.9 | 53.6 | 64.7 | 49.3 | -21.1 |
| FnCTOD (GPT-4 SOTA) | 58.8 | 45.2 | 63.2 | 69.5 | **76.4** | 62.6 | +4.2 |
| HiCoLoRA (LLaMA2-13B) | 62.0 | 42.0 | 61.0 | 65.0 | 69.0 | 60.0 | – |
| HiCoLoRA (FnCTOD Dataset) | **62.8** | **49.2** | **63.9** | **70.3** | 69.4 | **63.1** | +5.2 |

Table 5: Performance comparison between HiCoLoRA and FnCTOD under different settings on MultiWOZ (JGA %).

| Method | Attr. | Hotel | Rest. | Train | Taxi | AVG. |
|---|---|---|---|---|---|---|
| HiCoLoRA (Full) | **38.9** | **20.4** | **31.0** | **44.9** | **68.6** | **40.8** |
| w/ Swap Hier Strategies | 37.2 | 19.7 | 22.9 | 40.2 | 67.5 | 37.4 |
| w/o Adaptive Fusion | 28.9 | 19.3 | 20.3 | 43.0 | 68.0 | 35.9 |
| w/o Spec Joint Cluster | 36.2 | 19.8 | 27.5 | 42.1 | 63.6 | 37.8 |
| w/ Kiming Init | 34.3 | **20.4** | 27.8 | 40.4 | 67.5 | 38.1 |
| w/ PiSSA Init | 36.5 | 20.3 | 29.0 | 42.5 | 67.8 | 38.9 |
| w/ MiLoRA Init | 34.1 | 19.9 | 26.2 | 38.5 | 62.9 | 36.3 |

Table 6: Ablation study on hierarchical architecture, adaptive fusion, spectral clustering, initialization of HiCoLoRA on MultiWOZ. Attr. and Rest. are abbreviations for Attraction and Restaurant, respectively.

imental setup and efficiency. A thorough examination of FnCTOD's experimental configuration reveals several deviations from a strict zero-shot setting.

FnCTOD uses a carefully curated dataset of 7,200 dialogues across 36 domains (including SGD, CamRest676, MSR-E2E, TaskMaster, and WOZ), which include domains overlapping with MultiWOZ test domains. This violates the strict zero-shot learning premise. In contrast, HiCoLoRA uses only 4,625–7,684 samples from 4 domains in MultiWOZ, with one domain excluded during training to ensure a strict zero-shot setting. To ensure a fair comparison, we conducted an additional experiment by training FnCTOD on the **FnCTOD dataset**. The results, summarized in Table 5, demonstrate that FnCTOD achieves superior performance while maintaining significantly higher efficiency.

Beyond the fundamental discrepancy in training data composition, our comparative analysis reveals several critical distinctions that underscore HiCoLoRA's methodological rigor and practical efficiency: (1) When trained on identical data, HiCoLoRA achieves a JGA of 63.1, surpassing FnCTOD by 6.1% and even exceeding GPT-4-based FnCTOD by 0.5 JGA points; (2) HiCoLoRA maintains superior inference efficiency, requiring only a single LLM call with 16 token prompts versus FnCTOD's dual invocations and larger than 1200 token inputs; (3) While FnCTOD(without fine-tune) employs 5 few-shot examples in its zero-shot configuration (achieving only 49.3 JGA), HiCoLoRA operates under strict zero-shot conditions to attain 60.0 JGA; (4) FnCTOD's incorporation of detailed schema descriptions deviates from minimal prompt principles, whereas HiCoLoRA relies solely on its hierarchical adaptation mechanism; (5) Architecturally, HiCoLoRA achieves competitive performance through semantic aware initialization and efficient parameter updates, avoiding the computational overhead of prompt heavy approaches.

This comparative analysis demonstrates that FnCTOD not only achieves state-of-the-art performance under strict zero-shot settings but also offers superior efficiency and scalability compared to prompt heavy LLM-based approaches. The gains are attributable to its principled hierarchical adaptation, spectral semantic disentanglement, and knowledge preserving initialization mechanisms that are both empirically effective and practically efficient.

## A.5 ABLATION STUDY RESULTS

Table 6 validates the contributions and necessity of each core component of HiCoLoRA to its overall performance. This validation is conducted by systematically removing or replacing core com-

Table 7: Comparison with CAPID on MultiWOZ 2.1

| Method | Configuration | Attr. | Hotel | Rest. | Train | Taxi | AVG. |
|--------|---------------|-------|-------|-------|-------|------|------|
| CAPID | T5-base + T5-base | 40.9 | 43.5 | 37.1 | 49.5 | 87.1 | 50.1 |
| CAPID | T5-base + T5-small | 33.3 | 31.1 | 31.6 | 34.3 | 65.4 | 40.7 |
| HiCoLoRA (Ours) | T5-small | **38.9** | 20.4 | 31.0 | **44.9** | 68.6 | **40.8** |

ponents, including the hierarchical strategy, adaptive fusion, spectral clustering, and initialization method.

## A.6 COMPARISON WITH RECENT HETEROGENEOUS METHODS

To further validate the effectiveness of HiCoLoRA against contemporary approaches with different architectural paradigms, we conducted comparative analyses with two recently proposed state-of-the-art methods: LDST (Feng et al., 2023) and CAPID (Dong et al., 2024).

**Comparison with LDST (EMNLP 2023):** LDST proposes an Assembled Domain-Slot Instruction Generation approach for DST. This method generates diverse instruction samples by randomly combining different instruction and input templates during fine-tuning, thereby reducing the model's sensitivity to prompt variations. For example:

```
Instruction:
    Track the state of the slot <hotel-area> in the input dialogue.
Input:
    [USER] I need to book a hotel in the east that has 4 stars.
    [SYSTEM] I can help you with that. What is your price range?
    [domain] hotel, [slot] area, it indicates area or place of the hotel.
    This slot is categorical and you can only choose from the following
        available values: center, east, north, south, west.
    If the slot is not mentioned in the dialogue, just return NONE.
    So the value of slot <hotel-area> is
```

We performed comparative experiments on MultiWOZ 2.1 using the LLaMA-7B backbone for both methods. The results demonstrate that HiCoLoRA maintains 1.9% advantage over LDST (57.8 vs. 56.7 Average JGA). This performance gain, coupled with HiCoLoRA's parameter efficient design, further validates the effectiveness of our hierarchical collaborative architecture in capturing complex dialog state dependencies.

**Comparison with CAPID (EMNLP 2024):** CAPID proposes Context-aware Auto-prompting and Instruction-following Contrastive Decoding. This approach employs a two stage framework where a context-aware slot query generation method via auto-prompting which initially using GPT-4, aligns the gap between source and target domains. The generated prompts are used to train a T5-base student model to independently produce context-aware slot queries. During inference, the fine-tuned T5-base student model first generates the prompt, which is then used by the trained DST model (T5-base or T5-small) to predict slot values.

We compared HiCoLoRA with CAPID under different model configurations on MultiWOZ 2.1 (Table 7). HiCoLoRA shows a marginal advantage of 0.1% in Average JGA over the CAPID configuration (T5-base + T5-small). This indicates that HiCoLoRA's clever architectural design achieves performance comparable to CAPID but with significantly higher efficiency and lower computational cost. Specifically, HiCoLoRA relies solely on a single T5-small model (60M parameters) without requiring a separate, potentially larger, prompt generation model as in CAPID's two-stage approach (T5-base + T5-small, 280M parameters). Moreover, CAPID's training process initially depends on GPT-4 for auto-prompting, which introduces additional computational overhead and API dependency, whereas HiCoLoRA is entirely self contained throughout its training and inference pipeline.

**Discussion:** HiCoLoRA demonstrates distinct advantages over contemporary approaches. It surpasses the architectural efficiency of full fine-tuning methods like LDST through parameter effective LoRA adaptation, streamlines the multi-stage inference pipeline characteristic of CAPID via a unified hierarchical model, and offers enhanced scalability by natively accommodating multi-domain

| Experiment | Buses | Events | Flights | Media | Messaging | Music | Payment | Trains | AVG. |
|---|---|---|---|---|---|---|---|---|---|
| HiCoLoRA (Original) | 54.0 | 55.1 | 30.7 | 75.9 | 67.7 | 35.8 | 26.7 | 55.8 | 50.2 |
| HiCoLoRA (MultiWOZ→SGD) | 52.4 | 51.8 | 29.2 | 70.6 | 63.6 | 33.7 | 24.8 | 54.7 | 47.6 |

Table 8: Cross dataset generalization performance (JGA %) from MultiWOZ to SGD

| Experiment | Attr. | Hotel | Rest. | Train | Taxi | AVG. |
|---|---|---|---|---|---|---|
| HiCoLoRA (Original) | 38.9 | 20.4 | 31.0 | 44.9 | 68.6 | 40.8 |
| HiCoLoRA (SGD→MultiWOZ) | 37.0 | 19.0 | 30.4 | 43.1 | 64.5 | 38.8 |

Table 9: Cross dataset generalization performance (JGA %) from SGD to MultiWOZ

dialogues without external dependencies. This positions HiCoLoRA as an optimally balanced solution, delivering robust performance with markedly greater practical efficiency for dialogue state tracking.

## A.7 GENERALIZATION ANALYSIS

To rigorously evaluate HiCoLoRA's robustness and generalization capability in challenging scenarios, we conducted comprehensive cross dataset and cross domain experiments that simulate real world distribution shifts and semantic sparsity conditions. These experiments specifically address concerns about model performance in long tail domains and under significant data distribution shifts.

### A.7.1 CROSS DATASET EVALUATION

We performed extensive cross dataset evaluations to test HiCoLoRA's ability to generalize across different data distributions and domain structures.

**MultiWOZ to SGD Transfer:** Trained exclusively on all MultiWOZ domains and evaluated on the complete SGD test set, requiring adaptation to SGD's broader and unfamiliar service domains. As shown in Table 8, under this challenging setup, HiCoLoRA maintained an average JGA of 47.6%, representing only a 5.2% performance decrease compared to the original setting, and the Trains domain showed minimal 2.0% decline. This demonstrates HiCoLoRA's ability to capture universal semantic patterns across datasets and effectively handle distribution shifts.

**SGD to MultiWOZ Transfer:** Trained on SGD domains and evaluated on MultiWOZ, testing transfer from diverse but shallower domains to more complex dialogue structures. As shown in Table 9, when transferring from diverse but shallower SGD domains to the more complex Multi-WOZ, HiCoLoRA maintained an average JGA of 38.8%, a decrease of only 4.9% from the original performance. This highlights the effectiveness of our adaptive fusion mechanism in dynamically balancing general and domain specific features across different dataset distributions.

### A.7.2 LOW SEMANTIC OVERLAP TRANSFER

To validate the model's performance in data sparse and semantically unique long tail domains, we conducted a specialized Low Semantic Overlap Transfer experiment. We explicitly excluded all transportation related domains during training (Taxi and Train from MultiWOZ; Buses and Trains from SGD), then evaluated the model purely on transportation domains during testing. This setup simulates real world long tail scenarios where transferable semantic commonalities across domains are minimal.

Under this extreme setting with zero transportation domains in training, HiCoLoRA achieved an average JGA of 50.7% in transportation domains, a decrease of 9.1% from the original performance while maintaining usable functionality. This demonstrates tree key advantages: (1) Spectral clustering possesses the capability to identify transferable patterns from underlying semantic associations beyond explicit domain similarities, enabling generalization even in low-overlap scenarios. (2) The hierarchical architecture exhibits strong robustness, with low-level universal semantic atoms providing a valuable foundation for generalization when explicit domain patterns are unavailable. (3) The

| Experiment | Taxi (MultiWOZ) | Train (MultiWOZ) | Buses (SGD) | Trains (SGD) | AVG. |
|---|---|---|---|---|---|
| HiCoLoRA (Original) | 68.6 | 44.9 | 54.0 | 55.8 | 55.8 |
| HiCoLoRA (Cross-Dataset/Domain) | 62.8 | 38.8 | 49.7 | 51.3 | 50.7 |

Table 10: Low semantic overlap transfer performance (JGA %) in transportation domains

| Domain | Train | Dev | Test |
|---|---|---|---|
| Attraction | 2717 | 401 | 416 |
| Hotel | 3381 | 416 | 394 |
| Restaurant | 3813 | 438 | 207 |
| Taxi | 1654 | 207 | 195 |
| Train | 3103 | 484 | 494 |
| Total | 8438 | 1000 | 1000 |

Table 11: The dataset statistic of MultiWOZ.

| Domain | Train | Dev | Test |
|---|---|---|---|
| Buses | 2,280 | 329 | 526 |
| Events | 3,509 | 418 | 592 |
| Flights | 2,747 | 391 | 506 |
| Media | 1,113 | 179 | 364 |
| Messaging | NA | NA | 298 |
| Music | 1,290 | 196 | 347 |
| Payment | NA | NA | 222 |
| Trains | NA | NA | 350 |
| Total | 10,939 | 1,513 | 3,205 |

Table 12: The dataset statistic of SGD.

adaptive fusion mechanism offers dynamic flexibility, adjusting feature weights based on domain characteristics to avoid over reliance on specific domain patterns and maintain performance under distribution shifts.

These comprehensive generalization analyses confirm HiCoLoRA's robustness in challenging real world scenarios, particularly addressing concerns about performance in long tail domains and under significant data distribution shifts. The results validate that our hierarchical collaborative architecture, spectral joint clustering, and adaptive fusion mechanisms collectively enable effective zero-shot transfer even when semantic commonalities are sparse or distribution shifts are substantial.

## B  EXPERIMENTS SETTING DETAILS

### B.1  DATASET STATISTIC

Based on the experimental design for zero-shot dialog state tracking, domain selection was strategically constrained to ensure robust evaluation. For MultiWOZ (Table 11), the Police (46 dialogs) and Hospital (38 dialogs) domains were excluded due to insufficient dialog volume and slot diversity, which would compromise statistical reliability in zero-shot generalization tests. Similarly, in SGD (Table 12), services with limited samples or atypical slot structures, such as RideSharing (Test: 112), Calendar (Test: 98), etc., are omitted to avoid skew results. This curation focuses on evaluation on domains with adequate data density and representative slot semantics, ensuring that performance metrics reflect true zero-shot transferability rather than data-sparsity artifacts. Consequently, while coverage is reduced, the core challenge of cross-domain adaptation is preserved, with results generalizable to mainstream service-oriented interactions.

### B.2  BASELINE MODELS

In this section, we provide a detailed overview of each baseline, as outlined below.

#### B.2.1  MAIN BASELINE

- **TRADE** Wu et al. (2019) enhances dialog state generation by incorporating a copy mechanism and enabling knowledge transfer between tasks, allowing the model to handle unseen dialog states during training.
- **MA-DST** Kumar et al. (2020) leverages cross-attention to align context and slot representations across multiple semantic levels, while using self-attention on RNN hidden states to resolve cross-domain coreference.

- **SUMBT** Lee et al. (2019), built on the BERT-base, employs contextual semantic attention to learn the domain-slot-type and slot value relations, predicting slot values in a non-parametric manner.

- **SGD-baseline** Rastogi et al. (2019) encodes dialog history and schema elements using BERT and applies conditional prediction with schema embeddings to accommodate dynamic schema sets.

- **Seq2Seq-DU** Feng et al. (2021) formulates DST as a sequence-to-sequence task, using two BERT-based encoders to separately process dialog utterances and schema descriptions, followed by a pointer-based decoder to generate the dialog state.

- **GPT2-DST** Li et al. (2021a) utilizes a GPT2-base generative question answering model, enabling natural language queries to infer unseen constraints and slots for zero-shot generalization in multi-domain task-oriented dialogs.

- **TransferQA** Li et al. (2021b) integrates extractive and multiple-choice question answering within a unified text-to-text transformer framework, effectively tracking both categorical and non-categorical slots, and introducing unanswerable questions to improve robustness.

- **T5DST** Lin et al. (2021), based on T5-small and PPTOD-small, encodes dialog context and slot descriptions and generates slot values in an autoregressive manner. Slot-type descriptions facilitate cross-slot information sharing and cross-domain knowledge transfer.

- **SlotDM-DST** Wang et al. (2022), leveraging T5-small, models slot–slot, slot–value, and slot–context dependencies via slot prompts, value demonstrations, and constraint objects. Shared prompts capture transferable knowledge across domains.

- **Prompter** Aksu et al. (2023), based on PPTOD-small, generates dynamic prefixes from slot descriptions and injects them into the key and value states of each Transformer layer's self-attention mechanism, enabling zero-shot prefix tuning.

- **DCC** Wang et al. (2023) Divide, Conquer and Combine, built on T5-small, adopts a mixture-of-experts strategy by partitioning semantically independent data subsets, training corresponding experts, and applying ensemble inference for unseen samples.

- **DualLoRA** Luo et al. (2024) builds on PPTOD-small with a T5-small backbone, employing two low-rank adaptation matrices, one refining dialog context and the other slot prompts. Once trained, these matrices are fused into the frozen pre-trained weights, yielding zero-shot cross-domain dialog state tracking without any extra inference latency.

### B.2.2 LoRA Baseline

- **HydraLoRA** Tian et al. (2024) is a parameter-efficient fine-tuning (PEFT) framework designed to address the performance gap between standard LoRA and full fine-tuning, especially on complex datasets. Introduce an asymmetric LoRA structure that does not require domain expertise. Experiments demonstrate that HydraLoRA surpasses existing PEFT methods in performance.

- **LoRA-GA** Wang et al. (2024b) improves LoRA by proposing a novel gradient-aware initialization strategy that aligns the gradients of the low-rank matrices with those of full fine-tuning at the first training step. This method significantly accelerates convergence (2–4× faster than vanilla LoRA) and improves performance in tasks such as GLUE, GSM8K, and code generation, even for large models such as Llama 2-7B.

- **RoSA** Nikdan et al. (2024), Robust Adaptation combines low-rank and sparse adaptations inspired by robust PCA to approximate full fine-tuning performance under constrained computational budgets. It is particularly effective in generative tasks like math problem solving and SQL generation, and supports efficient training via custom sparse GPU kernels and compatibility with quantized base models.

- **Spectral Adapter** Zhang & Pilanci (2025) incorporates spectral information from pre-trained weights via SVD to enhance PEFT methods. Performs additive tuning or orthogonal rotation on the top singular vectors, improving rank capacity and parameter efficiency. The adapter also benefits multi-adapter fusion and demonstrates stronger performance across various tasks.

### B.2.3 LLM BASELINE

- **ChatGPT-zsTOD** Heck et al. (2023) achieves state-of-the-art performance in zero-shot dialog state tracking without task-specific training, leveraging its general-purpose language model capabilities. However, inherent limitations prevent it from fully replacing specialized systems, though its in-context learning abilities may support the development of dynamic dialog state trackers.

- **D0T** Finch & Choi (2024) enhances zero-shot DST by generating synthetic data across over 1,000 domains, creating a diverse training dataset with silver-standard annotations. This approach addresses data scarcity and enables adaptation to new domains without costly collection efforts.

- **MoPE** Tang et al. (2024) proposes a Mixture of Prefix Experts to connect similar slots across different domains, improving transfer performance in unseen domains. It addresses domain transferring and partial prediction problems in zero-shot DST.

- **FnCTOD** Li et al. (2024) improves zero-shot DST by calling functions with LLMs, allowing adaptation to diverse domains without extensive data or tuning. It achieves state-of-the-art performance with both open-source and proprietary LLMs, significantly boosting ChatGPT and GPT-4 results.

- **Multi-User** Song et al. (2025) evaluates LLMs in multi-user DST by extending datasets with second-user utterances generated via speech act theory. For a fair comparison, the experimental setup was configured using single-user data to evaluate the performance of LLMs in single-user dialog state tracking.

### B.3 EVALUATION METRIC FORMULAS

### B.3.1 JGA FORMULA

$$JGA = \frac{\sum_{i=1}^{T} I(S_i^{pre} = S_i^{gt})}{T} \tag{12}$$

In this formula, $T$ denotes the total number of dialog turns in the evaluation dataset. For each turn $i$, $S_i^{pre}$ and $S_i^{gt}$ represent the predicted and ground truth sets of slot-value pairs, respectively. The indicator function $I$ returns 1 if the inside condition is satisfied and 0 otherwise. Specifically, $I(S_i^{pre} = S_i^{gt})$ checks whether the predicted set of slot-value pairs for turn $i$ exactly matches the set of ground truth slot-value pairs. A value of 1 indicates a perfect match for that turn, that is, all slot value pairs were correctly predicted, while any discrepancy results in a value of 0. The summation $\sum_{i=1}^{T} I(S_i^{pre} = S_i^{gt})$ thus counts the number of turns for which the entire set of slot-value pairs was correctly predicted.

### B.3.2 AGA FORMULA

$$AGA = \frac{\sum_{i=1}^{T} \frac{|S_i^{gt} \cap S_i^{pre}| - |S_i^{pre} - S_i^{gt}|unique}{|Si^{gt}|}}{T} \tag{13}$$

In this formula, $T$ denotes the total number of dialog turns in the evaluation dataset. For each turn $i$, $S_i^{pre}$ and $S_i^{gt}$ represent the predicted and ground truth sets of slot-value pairs, respectively. The formula calculates the slot-level accuracy for each turn by:

- Computing the intersection $|S_i^{gt} \cap S_i^{pre}|$, which counts correctly predicted slot-value pairs

- Computing $|S_i^{pre} - S_i^{gt}|unique$, which counts incorrectly predicted slots (by extracting unique slot names from the difference set)

- Subtracting incorrect predictions from correct predictions

- Normalization by the total number of ground truth slot-value pairs $|Si^{gt}|$

The outer summation averages these per-turn accuracies across all dialog turns. Note that this is a more complex metric than simple slot matching, as it accounts for both missed slots and incorrect slot predictions while considering slot name uniqueness.

### B.4 Experiments Implementation Details

Our experimental setup, designed for a precise comparison with previous work, follows that of DualLoRA Luo et al. (2024). We use the T5-small architecture (6 encoder/decoder layers, 512 hidden dimension, 8 attention heads) as the backbone for HiCoLoRA, with a LoRA rank of 8 for low-rank adaptation, initialized from PPTOD-small checkpoints, consistent with observations in DualLoRA that PPTOD Su et al. (2022) is particularly suitable for prompt-tuning due to its pre-training objectives.

For spectral clustering, the number of domain clusters ($M$) and slot clusters ($N$) are set as 2 and 3 for MultiWOZ, with 2 and 4 specified for SGD. These configurations are determined by maximizing the silhouette coefficient.

Training configurations include a batch size of 8 with gradient accumulation every 8 steps, the AdamW optimizer (weight decay 0.01, learning rate 1e-4, no scheduler), a fixed random seed of 3407, and 5 training epochs (early stopping after 5 consecutive validation loss plateaus).

For hierarchical processing, we use a $\alpha = 50\%$ full collaboration ratio with higher layers and a semantic enhancement coefficient $\lambda = 0.5$ to modulate singular values in semantically enhanced SVD initialization.

The training and validation sets exclude target domain data, while the test set retains only target domain instances. All experiments were conducted on NVIDIA GeForce RTX 5080 GPUs.

## C  Case Study

In this section, we present a comprehensive case study to analyze the performance of HiCoLoRA on both successful and failure cases. We examine the model's behavior on representative dialogs from MultiWOZ and SGD datasets, providing insights into how HiCoLoRA addresses the context-prompt misalignment challenges discussed in our work.

### C.1  Successful Cases

#### C.1.1  Success Case 1

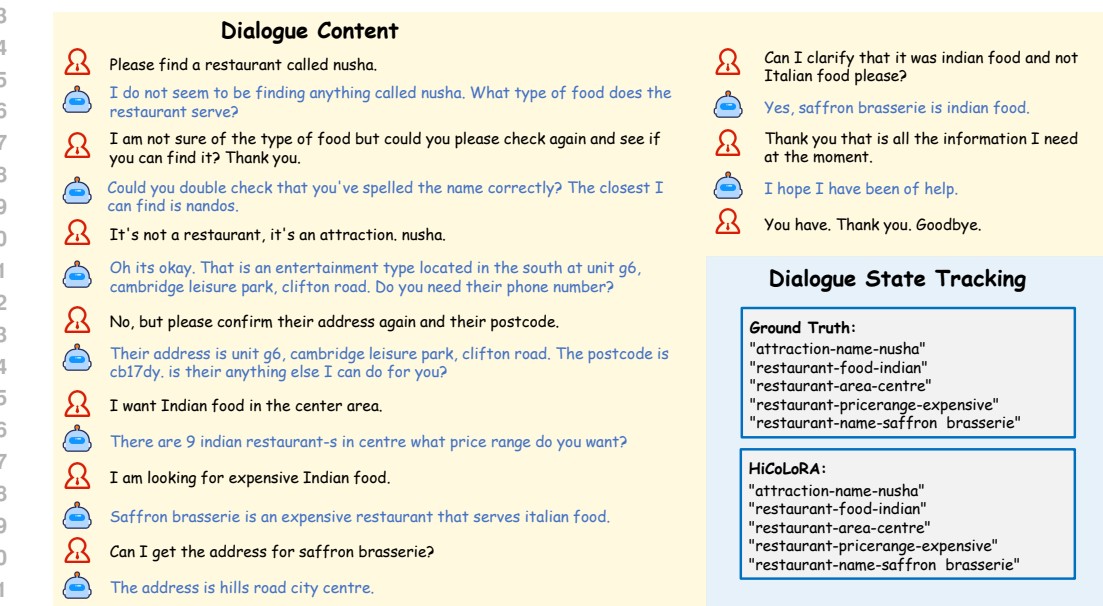

Figure 6: Success Case 1

**Dialog Context.** We analyze dialog PMUL4648.json (Fig. 6) from the MultiWOZ dataset where a user is seeking information about a restaurant named "saffron brasserie". The dialog involves mul-

tiple turns with complex slot-value interactions, including the restaurant name, food type (indian), price range (expensive), area (center).

**HiCoLoRA Performance.** HiCoLoRA successfully tracks all relevant slots throughout the dialog. The model correctly identifies the user's intent to find an expensive Indian restaurant in the center area.

**Analysis.** The success of HiCoLoRA in this case can be attributed to several factors:

1. **Hierarchical Collaboration:** The lower layers effectively capture local semantic features such as entity names and basic slot information, while the higher layers integrate these features to form a coherent understanding of the user's intent.

2. **Spectral Joint Clustering:** The model successfully identifies transferable domain-slot associations, enabling effective knowledge transfer between the attraction and restaurant domains.

3. **Adaptive Fusion:** The adaptive linear fusion mechanism dynamically balances the contributions of UniRep-LoRA and SemAdapt-LoRA, allowing the model to adjust to the specific requirements of each dialog turn.

### C.1.2 SUCCESS CASE 2

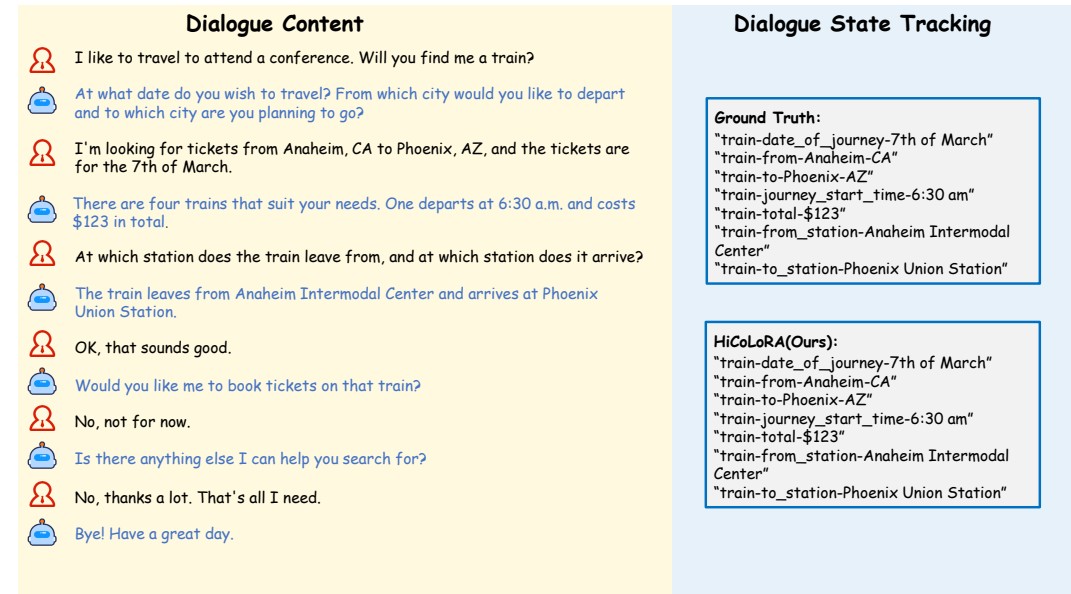

Figure 7: Success Case 2

**Dialog Context.** We examine the dialog "8_00066" (Fig. 7) from the SGD dataset, specifically the Trains domain. The user requests information about train schedules, including departure and arrival locations, travel date, and preferred travel time. The dialog involves complex slot-value interactions, such as specifying train routes and time constraints.

**HiCoLoRA Performance.** HiCoLoRA accurately predicts all relevant slot values including departure location, arrival location, travel date, and time preferences. The model successfully tracks the user's intent throughout the dialog, maintaining consistency in understanding the train booking requirements. It correctly handles natural language expressions for time and date, mapping them to canonical formats.

**Analysis.** The success in this SGD case demonstrates:

1. **Cross-Domain Generalization:** HiCoLoRA effectively generalizes to unseen domains in the SGD dataset, achieving high accuracy (55.8 JGA, 93.8 AGA) even in domains not encountered during training.

2. **Temporal Expression Handling:** The model successfully processes natural language temporal expressions and maps them to canonical time formats, which is crucial for train schedule queries.

3. **Semantic-Enhanced Initialization:** The SemSVD-Init mechanism preserves pre-trained knowledge, enabling the model to maintain performance on specialized domains with technical terminology, as evidenced by the high AGA.

## C.2 FAILURE ANALYSIS

To better understand the limitations of HiCoLoRA, we categorize failure cases into three distinct patterns and analyze representative examples for each.

### C.2.1 PATTERN 1: AMBIGUOUS SLOT BOUNDARY CASES

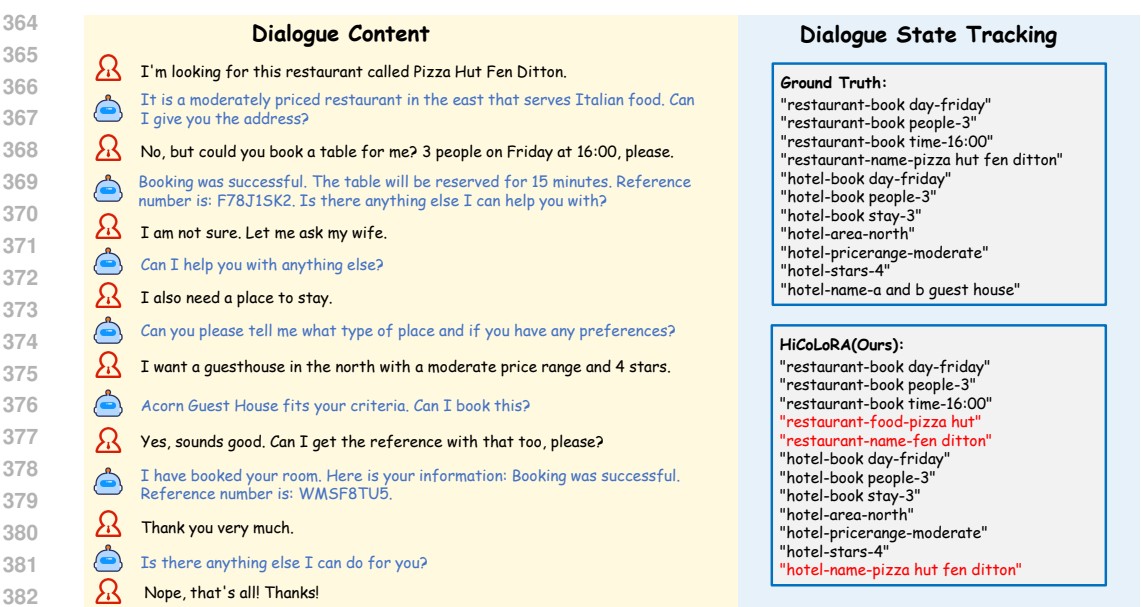

Figure 8: Failure Pattern 1: Ambiguous Slot Boundary Cases

**Description.** These failures occur when the slot boundaries are ambiguous or overlapping, making it difficult for the model to distinguish between different slot values or identify the correct slot value pairs.

**Example.** In MultiWOZ dialog PMUL4440.json (Fig. 8) involving both restaurant and hotel booking, HiCoLoRA exhibits significant prediction errors. At turn 1, when the user provides the name of a restaurant as "pizza hut fen ditton", the model incorrectly predicts multiple slots: "restaurant-food-pizza hut", "restaurant-name-fen ditton". Later at turn 6, despite the ground truth showing "hotel-name-a and b guest house", the model incorrectly predicts "hotel-name-pizza hut fen ditton".

**Analysis.** This type of failure highlights challenges in:

1. **Entity Recognition:** Distinguishing between different types of entities (area vs. parking) when they appear in close proximity in the user utterance.

2. **Implicit Slot Detection:** Recognizing implicitly mentioned slots that are not explicitly requested but are relevant to the user's intent.

### C.2.2 PATTERN 2: CROSS-DOMAIN CONFUSION

**Description.** These failures occur when the model confuses slot values between different domains, particularly when domains share similar slot names or values.

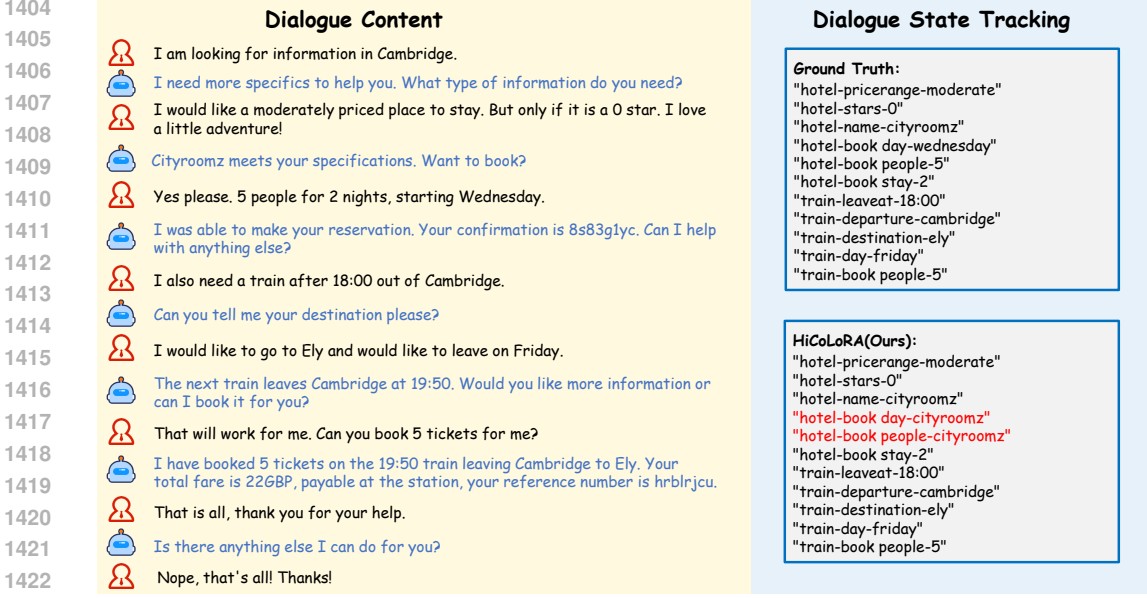

Figure 9: Failure Pattern 2: Cross-Domain Confusion

**Example.** In MultiWOZ dialog PMUL3514.json (Fig. 9), HiCoLoRA shows confusion in domain-specific slot value prediction. At turns 3-6, despite the ground truth consistently showing "hotel-name-cityroomz", the model incorrectly predicts "hotel-book day-cityroomz" and "hotel-book people-cityroomz", incorrectly associating the hotel name with booking slots. challenges in semantic entanglement even with disentanglement mechanisms.

**Analysis.** This failure pattern reveals limitations in:

1. **Domain Disambiguation:** Properly associating slot values with their respective domains in multi-domain dialogs.

2. **Contextual Understanding:** Maintaining clear separation between domain-specific contexts when processing complex multi-domain interactions.

3. **Semantic Overlap Handling:** Dealing with high-overlap domains where lexical similarities between slots from different domains cause confusion. This is particularly challenging when domain-agnostic features are overweighted by the adaptive fusion mechanism.

### C.2.3  PATTERN 3: RARE SLOT VALUE CASES

**Description.** These failures occur when the model encounters rare or unseen slot values that were not adequately represented in the training data. Analysis of the MultiWOZ and SGD datasets reveals that such slots are common: in *Attraction*, slots like "entrance fee" and "phone" appear in <10% of dialogs; in *Hotel*, "stars" and "internet" have fill rates <20%; in *Train*, "trainID" appears in <5% of dialogs. In a zero-shot setting, HiCoLoRA must generalize to both unseen domains and these rare slot values without any domain specific training examples, presenting a significant challenge.

**Example.** In MultiWOZ dialogs, HiCoLoRA struggles with predicting rare slot values for specific domains. For instance, in attraction domain dialogs, when users request detailed information about "entrance fee" or "address", the model often fails to correctly predict these values. Similarly, in hotel domain dialogs, when users inquire about specific details like "stars" or "internet", the model shows poor performance. In SGD dialogs, similar patterns emerge. For train domain dialogs, HiCoLoRA often fails to predict "trainID" or "price" information, particularly when these values are not explicitly mentioned in the user utterance but are expected as part of the system response.

**Analysis.** This failure pattern indicates challenges in:

1. **Rare Value Generalization:** Extending knowledge to handle infrequent slot values that may not have been adequately learned during pre-training. In a zero-shot setting, the model cannot benefit from domain-specific fine-tuning to improve performance on these rare slots.

2. **Contextual Inference:** Properly inferring rare slot values from contextual clues when they are not explicitly mentioned. This is particularly challenging for slots like "trainID" or "reference number" that require the model to generate specific identifiers.

3. **Domain-Aware Initialization:** Current initialization methods (SemSVD-Init) preserve pre-trained knowledge but may not adequately address domain-specific rare slot challenges. Future work could explore domain-aware initialization strategies that better account for rare slot distributions.

4. **Idiosyncratic Semantics Handling:** Dealing with slots that have domain-exclusive terms or idiosyncratic semantics that resist transfer. Spectral clustering may fail for slots with low-frequency terms, and semantic dilution in higher layers can occur when full collaboration fuses these slots with irrelevant ones.

## C.3 DISCUSSION

The case study analysis reveals both the strengths and limitations of HiCoLoRA. The successful cases demonstrate the effectiveness of our hierarchical collaborative architecture, spectral joint clustering, and semantic-enhanced initialization in addressing the core challenges of context-prompt misalignment. However, failure cases highlight areas for future improvement, particularly in handling ambiguous slot boundaries, cross-domain confusion, and rare slot values.

These findings suggest that, while HiCoLoRA represents a significant advance in zs-DST, more research is needed to address the identified failure patterns. Potential directions include:

1. **Enhanced Slot Boundary Detection:** Develop more sophisticated mechanisms to identify and separate slot boundaries in complex utterances.

2. **Improved Domain Disambiguation:** Exploring techniques for better domain separation in multi-domain dialogs.

3. **Rare Value Enhancement:** Investigating data enhancement strategies to improve coverage of rare slot values during training.

In general, the case study provides valuable insight into the practical performance of HiCoLoRA and informs future research directions on zs-DST.