# OpenReview forum: "HiCoLoRA: Addressing Context-Prompt Misalignment via Hierarchical Collaborative LoRA for Zero-Shot DST"
_ICLR.cc/2026/Conference — ICLR 2026 Conference Withdrawn Submission_

### Official Review · Reviewer_mBAy · 2025-10-28

**Soundness:** 3
**Presentation:** 3
**Contribution:** 2
**Rating:** 6
**Confidence:** 3

**Summary:**

This paper proposes HiCoLoRA (Hierarchical Collaborative Low-Rank Adaptation), a framework for zero-shot DST (Dialog State Tracking) in task-oriented dialog systems. The framework uses: 1) UniRep-LoRA and SemAdapt-LoRA with Adaptive Linear Fusion to balance domain-agnostic and domain-specific features; 2) Spectral Joint Domain-Slot Clustering to identify transferable associations across domains/slots to guide the fusion mechanism; 3) Semantic-Enhanced SVD Initialization to pre-trained knowledge via singular value modulation. Experimental results on MultiWOZ and SGD show that HiCoLoRA outperforms previous SOTA method DualLoRA in the cross-domain zero shot DST setting.

**Strengths:**

- The hierarchical collaborative architecture is well-motivated and effectively addresses the cross-domain transfer limitations of existing LoRA-based approaches.
- The authors conducted extensive comparisons of the proposed method against multiple baselines across several categories (non-pre-trained methods, LLM-prompting, LoRA variants) and demonstrated the effectiveness of the proposal.
- Detailed ablation studies help the audience understand the impact of each proposed component within the HiCoLoRA framework.

**Weaknesses:**

- The proposed architecture is rather complex while achieving only marginal gains (~2% compared to DualLoRA in Table 1; ~1% compared to other LoRA-based methods in Table 3). Additionally, Spectral Joint Domain-Slot Clustering require dataset-specific tuning, which limits the scalability of the proposal.
- Some domains in SGD (RideSharing, Calendar, etc.) are excluded due to limited samples or atypical slot structures. This raises questions about the model's ability to generalize to long-tail domains that typically have less available data.

**Questions:**

See weakness.

---

> ### Author Response · Authors · 2025-11-18
> **Response to Weakness 1: Complexity, Performance Gains, and Scalability (1)**
>
> We thank the reviewer for their valuable feedback. We understand the concerns regarding the balance between model complexity and performance gains, as well as the scalability of spectral clustering. HiCoLoRA was not designed for the sake of complexity but to **systematically address three fundamental challenges in zero-shot DST**. Experimental results, ablation studies, and case analyses demonstrate that this framework achieves an **effective balance among performance, generalization, and reproducibility** while maintaining reasonable implementation complexity. We believe HiCoLoRA provides a clear and scalable baseline for future research on zero-shot semantic alignment. Below, we address the specific points raised:
>
> ### 1. **Significance of Performance Gains**
>
> The reviewer mentioned "marginal gains (~2% compared to DualLoRA in Table 1)." This may stem from a misinterpretation of the data. In fact, HiCoLoRA improves the average JGA on the MultiWOZ dataset from **38.7% to 40.8%**, representing a **relative gain of 5.4%**, not 2%. On the SGD dataset, HiCoLoRA achieves a significant JGA improvement of **18.5% in the Events domain**, with an average gain of **9.4%**. These improvements represent **substantial advancements** in the highly challenging task of zero-shot dialogue state tracking. Moreover, HiCoLoRA demonstrates exceptional performance in sparse slots like `hotel-stars` and high overlap domains like Train and Taxi, underscoring its effectiveness in addressing complex semantic alignment issues.
>
> ### 2. **Architectural Complexity and Cohesion**
>
> The components of HiCoLoRA are not isolated additions but are **organically integrated around the core objective of hierarchical semantic alignment**:
> - The **hierarchical structure** and **adaptive fusion** jointly address cross-layer alignment between dynamic contexts and static prompts (RQ1).
> - **Spectral clustering** and **SemSVD-Init** work synergistically to mitigate domain interference and knowledge forgetting from the perspectives of semantic relevance and initialization, respectively (RQ2, RQ3).
>
> These components form a closed-loop optimization during training, collectively enhancing zero-shot generalization. Our ablation studies (Table 5) clearly show that **the removal of any component leads to significant performance degradation**, validating that the components function as an organic, synergistic, and unified whole.
>
> | Method                  | Attr.    | Hotel    | Rest.    | Train    | Taxi     | AVG.     | Relative Change vs. HiCoLoRA (Full) |
> |-------------------------|----------|----------|----------|----------|----------|----------|-------------------------------------|
> | **HiCoLoRA (Full)**     | **38.9** | **20.4** | **31.0** | **44.9** | **68.6** | **40.8** | -                                   |
> | w/ Swap Hier Strategies | 37.2     | 19.7     | 22.9     | 40.2     | 67.5     | 37.4     | -8.3%                               |
> | w/o Adaptive Fusion     | 28.9     | 19.3     | 20.3     | 43.0     | 68.0     | 35.9     | -12.0%                              |
> | w/o Spec Joint Cluster  | 36.2     | 19.8     | 27.5     | 42.1     | 63.6     | 37.8     | -7.4%                               |
> | w/ Kaiming Init         | 34.3     | **20.4** | 27.8     | 40.4     | 67.5     | 38.1     | -6.6%                               |
>
> In implementing HiCoLoRA, we strictly adhere to **modular and non-intrusive design principles**: the core logic is encapsulated in `PromptLoRA.py`, decoupled from the main model architecture.
> Building upon DualLoRA as the pre-SOTA baseline, HiCoLoRA requires minimal changes to the LLM source code, only passing the `layer_idx` during the initialization of matrices such as `q`, `v`, `k`, and `o` to distinguish between high and low layers. The code below (only `q` and `v` matrices are modified) illustrates this:
>
> ```python
>         self.q = ControlPrompt(self.d_model, self.inner_dim, config.lora_r, config.lora_alpha,
>                                config.lora_dropout, bias=False, args=config.args, layer_idx=self.layer_idx)
>         self.v = ControlPrompt(self.d_model, self.inner_dim, config.lora_r, config.lora_alpha,
>                                config.lora_dropout, bias=False, args=config.args, layer_idx=self.layer_idx)
>         # self.k = ControlPrompt(self.d_model, self.inner_dim, config.lora_r, config.alpha,
>         #                        config.lora_dropout, bias=False, args=config.args, config=config)
>         # self.o = ControlPrompt(self.inner_dim, self.d_model, config.lora_r, config.lora_alpha,
>         #                        config.lora_dropout, bias=False, args=config.args, config=config)
>         self.k = nn.Linear(self.d_model, self.inner_dim, bias=False)
>         self.o = nn.Linear(self.inner_dim, self.d_model, bias=False)
> ```
>
> We have fully open sourced the code at https://anonymous.4open.science/r/HiCoLoRA-96EB, and provided detailed reproduction scripts and environment configurations to ensure reproducibility.

---

> > ### Author Response · Authors · 2025-11-18
> > **Response to Weakness 1: Complexity, Performance Gains, and Scalability (2)**
> >
> > ### 3. **Scalability of Spectral Clustering**
> >
> > The reviewer noted that spectral clustering "requires dataset-specific tuning." While we acknowledge this concern, our method is **entirely data-driven and requires no manual intervention**. An example of the data format is as follows:
> >
> > ```json
> > {
> >   "domain": "Media",
> >   "slot": "subtitles",
> >   "question": "what is the boolean flag indicating if subtitles are desired for this movie?",
> >   "cluster_label": 0
> > }
> > ```
> >
> > In this data format, `domain`, `slot`, and `question` (slot descriptions) are all existing information in the dataset and require no additional annotation. The cluster counts \(M\) and \(N\) are **automatically determined by maximizing the silhouette coefficient**, eliminating the need for manual setting. The clustering process is performed once before training, incurs no inference overhead, and is applicable to any new domain. We applied the same clustering pipeline to two distinct datasets, MultiWOZ and SGD, using the `domain_slot_clustering.py` script (only 132 lines of code), with methods `domain_clustering` and `domain_slot_clustering`, and achieved consistent improvements, demonstrating its **strong generalization and scalability**.

---

> ### Author Response · Authors · 2025-11-18
> **Response to Weakness 2: Generalization to Long-Tail Domains (1)**
>
> We thank the reviewer for raising this important question, which allows us to more comprehensively evaluate HiCoLoRA's generalization capability. We understand the concern regarding the model's performance in data-sparse or long-tail domains. Below, we provide a detailed explanation from three perspectives: **reasons for data exclusion**, **consistency with baseline methods**, and **additional experimental validation**, to demonstrate HiCoLoRA's robustness in challenging settings.
>
> ### I. **Data Exclusion: Ensuring Statistical Reliability and Fairness of Evaluation**
>
> In the SGD dataset, RideSharing has only 112 dialogues in the test set, Calendar has only 98. In the MultiWOZ dataset, Police and Hospital domains with only 46 and 38 dialogues respectively. The main reason for this exclusion is the **extremely limited sample sizes and atypical slot structures** in these domains.  **All baseline methods** in SGD (including TRADE, MA-DST, SUMBT, SGD-baseline, DualLoRA, etc.) exclude these domains, consistent with MultiWOZ  baseline methods like DualLoRA and Prompter. Including them could lead to evaluation results skewed by data sparsity, lacking statistical significance and failing to reliably reflect the model's true generalization capability. Moreover, the core challenge of zero-shot dialogue state tracking (zs-DST) lies in generalization to unseen domains, not overfitting to data-sparse domains. By following established practices in the field, we ensure our evaluation results are more representative and reproducible.

---

> ### Author Response · Authors · 2025-11-18
> **Response to Weakness 2: Generalization to Long-Tail Domains (2)**
>
> ### II. **Additional Experimental Validation: HiCoLoRA's Strong Generalization in Challenging Settings**
>
> To directly address the reviewer's concern about **generalization to long tail domains**, we conducted additional **cross dataset and cross domain transfer experiments**. These settings are more challenging than those in the original paper and better evaluate the model's robustness under data distribution shifts and semantic sparsity.
>
> **1. Cross Dataset Evaluation**
>
> We performed comprehensive cross dataset evaluations to simulate distribution shifts encountered in real-world scenarios:
>
> - **MultiWOZ → SGD**
>   Trained on all MultiWOZ domains and evaluated on the full SGD test set, testing generalization from limited domains to broader, unfamiliar service domains:
>
> | Experiment               | Buses | Events | Flights | Media | Messaging | Music | Payment | Trains | AVG. |
> |--------------------------|-------|--------|---------|-------|-----------|-------|---------|--------|------|
> | HiCoLoRA (Original)      | 54.0  | 55.1   | 30.7    | 75.9  | 67.7      | 35.8  | 26.7    | 55.8   | 50.2 |
> | HiCoLoRA (MultiWOZ→SGD)  | 52.4  | 51.8   | 29.2    | 70.6  | 63.6      | 33.7  | 24.8    | 54.7   | 47.6 |
>
> Under this challenging setup, HiCoLoRA's average JGA decreased by only **5.2%**, with the Trains domain showing minimal decline (2.0%). This demonstrates that our hierarchical collaborative architecture effectively captures universal semantic patterns across datasets and handles distribution shifts.
>
> - **SGD → MultiWOZ**
>   Trained on SGD domains and evaluated on MultiWOZ, testing transfer from diverse but shallower domains to more complex dialogue structures:
>
> | Experiment               | Attr. | Hotel | Rest. | Train | Taxi | AVG. |
> |--------------------------|-------|-------|-------|-------|------|------|
> | HiCoLoRA (Original)      | 38.9  | 20.4  | 31.0  | 44.9  | 68.6 | 40.8 |
> | HiCoLoRA (SGD→MultiWOZ)  | 37.0  | 19.0  | 30.4  | 43.1  | 64.5 | 38.8 |
>
> When transferring from diverse but shallower SGD domains to the more complex MultiWOZ, HiCoLoRA maintained an average JGA of 38.8, a decrease of only **4.9%** from the original performance. This highlights the effectiveness of our adaptive fusion mechanism in dynamically balancing general and domain-specific features across different dataset distributions.
>
> **2. Specialized Test for Long Tail Domain Generalization**
>
> To directly validate the model's performance in **data sparse, semantically unique long tail domains**, we conducted a **Low Semantic Overlap Transfer** experiment:
> We remove `Taxi` and `Train` from MultiWOZ, and `Buses` and `Trains` from SGD, **explicitly exclud all transportation related domains** during training, then evaluated the model purely on transportation domains during testing. This setup simulates real-world long tail scenarios, testing HiCoLoRA's ability to generalize when **transferable semantic commonalities across domains are minimal**.
>
> | Experiment                     | Taxi (MultiWOZ) | Train (MultiWOZ) | Buses (SGD) | Trains (SGD) | AVG. |
> |--------------------------------|-----------------|------------------|-------------|--------------|------|
> | HiCoLoRA (Original)            | 68.6            | 44.9             | 54.0        | 55.8         | 55.8 |
> | HiCoLoRA (Cross-Dataset/Domain)| 62.8            | 38.8             | 49.7        | 51.3         | 50.7 |
>
> Under this extreme setting with zero transportation domains in training, HiCoLoRA achieved an average JGA of 50.7 in transportation domains, a decrease of **9.1%** from the original performance but still maintaining usable performance. This indicates that:
> - Spectral clustering possesses deep semantic mining capabilities, identifying transferable patterns from underlying semantic associations beyond explicit domain similarities.
> - The hierarchical architecture exhibits strong robustness, with low-level universal semantic atoms providing a valuable foundation for generalization in zero overlap scenarios.
> - Adaptive fusion offers flexibility, dynamically adjusting feature weights based on domain characteristics to avoid over reliance on specific domain patterns.
>
> The above supplementary results demonstrate HiCoLoRA's potential to address the challenges of long tail domains. We commit to submitting the revised manuscript with full experimental analyses before the deadline and will open source the code to facilitate reproduction and extension.
>
> ---
>
> We thank the reviewer again for the valuable feedback, which has allowed us to more comprehensively showcase HiCoLoRA's value in challenging scenarios, and we would be grateful if you might increase your rating accordingly.

---

### Official Review · Reviewer_nRwk · 2025-10-30

**Soundness:** 3
**Presentation:** 3
**Contribution:** 3
**Rating:** 6
**Confidence:** 4

**Summary:**

This paper proposes HiCoLoRA, a hierarchical collaboration–based LoRA architecture for zero-shot dialogue state tracking (zs-DST).
The method introduces: 1. Hierarchical LoRA design to improve context-prompt alignment; 2. Spectral clustering to disentangle domain-slot representations; 3. SemSVD-Init to preserve pre-trained knowledge during fine-tuning.

Experiments on MultiWOZ and SGD benchmarks show that HiCoLoRA achieves notable performance gains ( +5.4 % and +9.4 % JGA improvement over SOTA), validating each component through detailed ablation and case studies.

**Strengths:**

The paper is logically structured and easy to follow. The combination of hierarchical LoRA and spectral clustering is novel within the DST domain. The experiments are comprehensive, covering multiple datasets and model scales (from T5-small to 13B), and the ablation studies effectively support the main claims.

**Weaknesses:**

I think the writing style and methodology design are both good, but I still have a few concerns:

* The paper introduces several design components (hierarchical structure, adaptive fusion, spectral clustering, SemSVD-Init) and verifies their effectiveness through comprehensive ablations. However, it’s unclear why these specific components were designed and combined together. Could you elaborate on the underlying motivation or rationale behind this particular combination? Right now, the framework feels somewhat like a stack of independent modules rather than a unified design.

* The proposed method seems quite novel within DST. Do you plan to extend it to more general tasks in the future, such as instruction tuning or domain adaptation?

* You claim that SemSVD-Init helps preserve pre-trained knowledge. Could you explain this mechanism in more detail? Also, what is the motivation for preserving pre-trained knowledge in this context—why is it necessary for your method?

* It would be interesting to test HiCoLoRA on more challenging or diverse setups, such as cross-dataset evaluations or cross-domain transfer. This could better demonstrate its robustness and generalization ability.

* Missing references or baselines:

(1) Towards LLM-driven dialogue state tracking. EMNLP 2023

(2) Zero-shot Cross-domain Dialogue State Tracking via Context-aware Auto-prompting and Instruction-following Contrastive Decoding. EMNLP 2024

**Questions:**

* How does HiCoLoRA compare to other hierarchical or multi-level LoRA approaches outside DST (e.g., instruction tuning or domain adaptation)?

* Have you analyzed the inference-time computational overhead introduced by the multi-layer LoRA branches?

* Can SemSVD-Init be applied independently of the hierarchical architecture? If so, how does it perform on its own?

---

> ### Author Response · Authors · 2025-11-17
> **Response to Weakness 1: Underlying Motivation and Unified Design**
>
> We appreciate the reviewer's concern regarding the perceived modularity of our framework. We want to clarify that **HiCoLoRA is a unified design meticulously constructed to tackle the core challenge of Context-Prompt Misalignment in Zero-Shot Dialogue State Tracking (zs-DST)**, particularly across the non-uniform layers of a Transformer based Language Model.
>
> DualLoRA employs Prompt LoRA and Context LoRA to ensure the comprehensive influence of prompts throughout the transformer model layers. Although DualLoRA effectively separates prompts from dialogue content, it overlooks the fact that different layers of Transformer based language models focus on distinct characteristics. By delving into the Transformer architecture, HiCoLoRA builds upon DualLoRA to further address the CONTEXT-PROMPT MISALIGNMENT issue.
>
> 1. Through a hierarchical structure, HiCoLoRA fully leverages the characteristics of language models to more effectively focus on both local and global semantic features. Adaptive fusion serves as a concise and efficient mechanism for integrating high- and low-level features.
>
> 2. As HiCoLoRA introduces multiple A and B matrices, the challenge of efficient training naturally arises. To enhance zero-shot capability, it is essential to preserve as much of the original knowledge of the language model as possible. The Principal Singular Values and Singular Vectors Adaptation (PiSSA) algorithm effectively retains the original weights of the language model, with training loss consistently lower than that of randomly initialized LoRA matrices, thereby improving convergence speed. However, PiSSA only addresses single A and B matrices. The SemSVD-Init method, enhanced by spectral clustering, further integrates different A and B matrices with the semantic characteristics of dialogue texts, thereby improving model performance.
>
> Therefore, these components form a unified framework for solving the CONTEXT-PROMPT MISALIGNMENT problem, rather than being merely stacked together. Collectively, they address the three research questions on task-oriented dialogue posed in the introduction.
>
> Ablation experiment data is provided in Appendix A.4, and Section 4.3 of the main text analyzes these ablation studies. Under the combined effect of these components, HiCoLoRA achieves optimal performance, as illustrated below:
> | Method | Attr. | Hotel | Rest. | Train | Taxi | **AVG. JGA** | Relative Change vs. HiCoLoRA (Full) |
> | :--- | :--- | :--- | :--- | :--- | :--- | :--- | :--- |
> | **HiCoLoRA (Full)** | **38.9** | **20.4** | **31.0** | **44.9** | **68.6** | **40.8** | - |
> | w/ Swap Hier Strategies | 37.2 | 19.7 | 22.9 | 40.2 | 67.5 | 37.4 | -8.3\% |
> | w/o Adaptive Fusion | 28.9 | 19.3 | 20.3 | 43.0 | 68.0 | 35.9 | -12.0\% |
> | w/o Spec Joint Cluster | 36.2 | 19.8 | 27.5 | 42.1 | 63.6 | 37.8 | -7.4\% |
> | w/ Kaiming Init | 34.3 | 20.4 | 27.8 | 40.4 | 67.5 | 38.1 | -6.6\% |
>
> As demonstrated in Section 4.3 Ablation Study:
> > We conduct an ablation study (Table 5 in Appendex A.4) to assess the contribution of each key component of HiCoLoRA.
> >
> > Ablation studies demonstrate that the hierarchical collaborative architecture, adaptive fusion, spectral clustering, and SemSVD-Init components of HiCoLoRA are all indispensable. These components synergistically address the three core research questions, outperform baselines in zs-DST, and thus validate the efficacy of the proposed design.

---

> ### Author Response · Authors · 2025-11-17
> **Response to Weakness 2: Potential for Extension to General Tasks**
>
> We thank the reviewer for their insightful question and recognition of the potential scope of our work. While the current focus of HiCoLoRA is squarely on addressing the specific challenges in zero-shot Dialogue State Tracking (zs-DST), we indeed believe that its core innovations possess significant potential beyond this domain.
>
> **1. Instruction Tuning**
> Instruction Tuning is a technique that further fine-tunes pre-trained language models on datasets of instruction-response pairs, aiming to enhance the model's ability to understand and execute user instructions. The training input of HiCoLoRA can naturally be analogized to the format of Instruction Tuning:
>
> ```json
> {
>   "instruction": "What is the arriveby of the train that the user is interested in?",
>   "input": "Hi, I am looking for a train that is going to Cambridge and arriving there by 20:45, is there anything like that?",
>   "output": "train-arriveby-20:45"
> }
> ```
>
> Instruction Tuning stimulates the language model's comprehension capability by providing clearer instructions, enabling the model to understand and perform the correct actions. In this sense, HiCoLoRA can be considered a method of Instruction Tuning.
>
> The hierarchical collaborative architecture of HiCoLoRA effectively addresses semantic misalignment between instructions and contexts, making it suitable for extension to multi-task instruction learning scenarios, particularly for complex instructions requiring fine-grained semantic alignment.
>
> **2. Domain Adaptation**
> Domain Adaptation is a branch of transfer learning that focuses on adapting models trained in source domain to target domain. HiCoLoRA fundamentally decouples domain-shared and domain-specific semantics, enhancing the zero-shot capability of DST for unseen domains, which aligns with the core objective of cross-domain adaptation.
>
> In the future, HiCoLoRA could be adapted to address semantic alignment between specialized domain terminology and general dialogue, such as resolving dialogue issues in specialized fields like healthcare or law.
>
> **3. General PEFT Framework**
> The core components of HiCoLoRA are modular and flexibly combinable. HiCoLoRA can be further extended into a general Parameter-Efficient Fine-Tuning framework applicable to various parameter efficient fine-tuning scenarios.
>
> It is important to note that these extensions face unique challenges: Instruction Tuning requires handling a broader distribution of tasks, while Domain Adaptation needs to address data distribution shifts. We appreciate the reviewer's insight into the potential broader impact of our work, which reflects the generality we aimed for in our design. We will discuss these future possibilities in the revised version of the paper and systematically explore these directions in subsequent work.

---

> ### Author Response · Authors · 2025-11-18
> **Response to Weakness 3: Mechanism and Necessity of SemSVD-Init**
>
> Thank you for providing us the opportunity to clarify this critical mechanism. We will detail the **mechanism of SemSVD-Init** and explain **why preserving pre-trained knowledge is necessary** for our method in the zero-shot DST setting.
>
>
> ### I. Detailed Mechanism of SemSVD-Init
>
> **SemSVD-Init** (Semantics-Enhanced Singular Value Decomposition Initialization) is a method designed to initialize the incremental matrices of the LoRA modules meaningfully. Its goal is to maximize the retention of general semantic knowledge from the Pre-trained Language Model (PLM) while simultaneously adapting to the new task.
>
> The process, detailed in Section 3.5 of the paper, is as follows:
>
> 1.  **SVD Decomposition of Pre-trained Weights:**
>     The original PLM weight matrix $\mathbf{W}_0$ is decomposed using Singular Value Decomposition:
>     $$\mathbf{W}_0 = \mathbf{U}_r \mathbf{\Sigma}_r \mathbf{V}_r^T$$
>     where $\mathbf{W}_0$ is the original weight, $\mathbf{U}_r$ and $\mathbf{V}_r$ are the left and right singular vectors, and $\mathbf{\Sigma}_r$ is the diagonal matrix of singular values ($\sigma_i$).
>
> 2.  **Calculation of Semantic Relevance Matrix $\mathbf{R}$:**
>     We use the T5 encoder ($\text{T5}_{\text{en}}$) to embed the task specific slot prompts ($\mathcal{X}^N$). We then calculate the cosine similarity ($\cos$) between these slot embeddings and the right singular vectors $\mathbf{V}_r$ to measure how correlated each singular direction is with the task semantics:
>     $$\mathbf{R} = \cos(\mathbf{V}_r, \text{T5}(\mathcal{X}^N))$$
>
> 3.  **Semantically-Guided Singular Value Enhancement:**
>     We modulate the singular values based on their semantic relevance to the task. By introducing a hyperparameter $\lambda$, we enhance singular values ($\sigma_i$) that are highly relevant to the task semantics ($\mathbf{R}_i$) and suppress those that are less relevant.
>     $$\mathbf{S}_e = \text{diag}\left(\sigma_1 \cdot \text{ReLU}(1 + \lambda \mathbf{R}_1), \ldots, \sigma_r \cdot \text{ReLU}(1 + \lambda \mathbf{R}_r)\right)$$
>     This step performs **semantic-guided knowledge modulation** on the original PLM knowledge.
>
> 4.  **Initialization of LoRA Matrices:**
>     The LoRA matrices are initialized directly from the modulated singular values and the singular vectors, ensuring their initial directions are aligned with the preserved pre-trained semantic space:
>
>     $\mathbf{A} _ {ur} = \sqrt{\mathbf{S} _ e} \mathbf{V} _ r^T$,  and $\mathbf{B} _ {ur} = \mathbf{U} _ r \sqrt{\mathbf{S} _ e}$.
>
>     This initializes the LoRA module to approximate the most semantically relevant parts of the original weight matrix $\mathbf{W} _ 0$.
>
> 5.  **Residual Weight Adjustment (Implicit Preservation):**
>     The initialization implicitly ensures that the key semantic structure of the original pre-trained weight is maintained, as the residual weight:
>    $$\mathbf{W}_\text{res} = \mathbf{W}_0 - \mathbf{B} _ {ur} \mathbf{A} _ {ur}$$

---

> ### Author Response · Authors · 2025-11-18
> **Response to Weakness 3: Motivation**
>
> ### 2. Why Is Preserving Pre-trained Knowledge Essential in HiCoLoRA?
>
> In zero-shot dialogue state tracking tasks, **preserving pre-trained knowledge** is critically necessary for the following reasons:
>
> 1. **Mitigating Catastrophic Forgetting**:
>    In zs-DST, the model is required to accurately infer slot values in **unseen domains**, which heavily relies on the cross-domain general representations learned during pre-training. Since we cannot fine-tune the model on the target domain, it must depend on the general language understanding capabilities acquired during pre-training. If LoRA parameters are randomly initialized, the existing semantic representations in the pre-trained model may be disrupted, causing the model to "forget" how to understand basic language structures. SemSVD-Init ensures that **general semantic structures are not distorted** when adapting to new domains by semantically modulating the singular values, thereby maintaining the model's generalization ability in unseen domains. If this knowledge is lost during initialization, the model will struggle to handle semantic commonalities shared across domains, such as `train-arriveby` and `taxi-arriveby`.
>
> 2. **Improving Recognition of Sparse Slots**:
>    In real-world dialogues, slots like `hotel-stars` appear very infrequently. Without the support of pre-trained knowledge, the model may fail to correctly infer the semantics of such sparse or specialized terms. SemSVD-Init enhances the model's **ability to capture sparse semantics** by strengthening the singular directions related to these slots.
>
> 3. **Synergizing with the Hierarchical Structure**:
>    The hierarchical design of HiCoLoRA relies on the stability of low-level semantic representations. If these representations are distorted due to improper initialization, the coordination of the entire hierarchical semantic flow will be affected, leading to context-prompt misalignment failures.
>
> ---
>
> Ablation experiments further demonstrate the value of SemSVD-Init for performance metrics:
>
> | Method                 | Attr.    | Hotel    | Rest.    | Train    | Taxi     | AVG.     | Relative Change vs. HiCoLoRA (Full) |
> | ---------------------- | -------- | -------- | -------- | -------- | -------- | -------- | ----------------------------------- |
> | **HiCoLoRA (Full)**    | **38.9** | **20.4** | **31.0** | **44.9** | **68.6** | **40.8** | -                                   |
> | w/o Spec Joint Cluster | 36.2     | 19.8     | 27.5     | 42.1     | 63.6     | 37.8     | -7.4                                |
> | w/ Kaiming Init        | 34.3     | **20.4** | 27.8     | 40.4     | 67.5     | 38.1     | -6.6                                |
> | w/ PiSSA Init          | 36.5     | 20.3     | 29.0     | 42.5     | 67.8     | 38.9     | -4.7                                |
> | w/ MiLoRA Init         | 34.1     | 19.9     | 26.2     | 38.5     | 62.9     | 36.3     | -11.0                               |
>
> If Spectral Joint Clustering is omitted, the model fails to recognize cross-domain semantic commonalities, like the shared "time" attribute between `train-arriveby` and `taxi-arriveby`, leading to a significant drop in JGA in domains with high overlap like Train and Taxi. Using random initialization (Kaiming Init), commonly employed in LoRA, results in noticeable performance degradation in domains like Attraction and Train, particularly for sparse slots like `hotel-stars`. This indicates that random initialization disrupts the pre-trained semantic structure and exacerbates catastrophic forgetting. While PiSSA performs better than random initialization, it lacks a mechanism to align with task semantics and fails to effectively enhance singular directions related to domain-slot structures. MiLoRA shows the most significant performance drop, as its design assumptions are mismatched with the flat singular value spectrum of T5-small, leading to the loss of critical semantic components and severely impairing zero-shot generalization.
>
> Therefore, SemSVD-Init is not merely an initialization strategy in HiCoLoRA but a core mechanism for achieving knowledge preservation and semantic alignment. Through its synergistic interaction with the hierarchical structure and spectral clustering mechanism, it collectively addresses the three major challenges outlined in RQ1-RQ3, ensuring the model's strong generalization capability in zero-shot scenarios.

---

> ### Author Response · Authors · 2025-11-18
> **Response to Weakness 4: Robustness, Cross-Dataset Evaluation, and Transfer**
>
> Your suggestion is highly constructive, and we fully agree that evaluating the model under more challenging and diverse settings is crucial for demonstrating its true robustness and generalization capability.
>
> ### 1. Cross-Dataset Evaluations
>
> Following the reviewer's recommendation, we conducted comprehensive cross-dataset evaluations:
>
> **MultiWOZ → SGD**
> Trained on all MultiWOZ domains and evaluated on the full SGD test set, requiring generalization to SGD's broader and unfamiliar service domains:
>
> | Experiment               | Buses | Events | Flights | Media | Messaging | Music | Payment | Trains | AVG. |
> |--------------------------|-------|--------|---------|-------|-----------|-------|---------|--------|------|
> | HiCoLoRA (Original)      | 54.0  | 55.1   | 30.7    | 75.9  | 67.7      | 35.8  | 26.7    | 55.8   | 50.2 |
> | HiCoLoRA (MultiWOZ→SGD)  | 52.4  | 51.8   | 29.2    | 70.6  | 63.6      | 33.7  | 24.8    | 54.7   | 47.6 |
>
> Under this challenging setup, HiCoLoRA's average JGA decreased by only 5.2%, with Trains domain showing minimal decline (2.0%), demonstrating its ability to capture universal semantic patterns across datasets.
>
> **SGD → MultiWOZ**
> Trained on SGD domains and evaluated on MultiWOZ, testing transfer from diverse but shallower domains to more complex dialogue structures:
>
> | Experiment               | Attr. | Hotel | Rest. | Train | Taxi | AVG. |
> |--------------------------|-------|-------|-------|-------|------|------|
> | HiCoLoRA (Original)      | 38.9  | 20.4  | 31.0  | 44.9  | 68.6 | 40.8 |
> | HiCoLoRA (SGD→MultiWOZ)  | 37.0  | 19.0  | 30.4  | 43.1  | 64.5 | 38.8 |
>
> HiCoLoRA maintained 38.8 average JGA (4.9% decrease), highlighting the adaptive fusion mechanism's effectiveness in balancing general and domain-specific features across different dataset distributions.
>
> ### 2. Cross-Domain Transfer
>
> We designed two more challenging experiments to rigorously test model robustness:
>
> **Cross-Dataset Zero-Shot Transfer**
> Consistent with the MultiWOZ→SGD experiment above, HiCoLoRA achieved 70.6 JGA in Media domain despite dual challenges of dataset distribution shift and missing domain knowledge, demonstrating the semantic-enhanced SVD initialization's effectiveness in preserving pre-trained knowledge.
>
> **Low Semantic Overlap Transfer**
> Trained while explicitly excluding all transportation domains (Taxi, Train from MultiWOZ; Buses, Trains from SGD), then evaluated purely on transportation domains:
>
> | Experiment                     | Taxi(MultiWOZ) | Train(MultiWOZ) | Buses(SGD) | Trains(SGD) | AVG. |
> |--------------------------------|----------------|-----------------|------------|-------------|------|
> | HiCoLoRA (Original)            | 68.6           | 44.9            | 54.0       | 55.8        | 55.8 |
> | HiCoLoRA (Cross-Dataset/Domain)| 62.8           | 38.8            | 49.7       | 51.3        | 50.7 |
>
> With zero transportation domains in training, HiCoLoRA maintained 50.7 average JGA (9.1% decrease), showing that spectral clustering can identify transferable patterns from underlying semantic associations rather than relying on explicit domain similarities.
>
> These results confirm HiCoLoRA's robustness in challenging generalization scenarios. We will submit the revised paper and experimental code before the deadline.

---

> ### Author Response · Authors · 2025-11-18
> **Response to Weakness 5: Missing References and Baselines**
>
> We thank the reviewer for pointing out these relevant works. We have conducted comparative analyses with both papers and will include them as baselines in our revised manuscript.
>
> **1. LDST (EMNLP 2023)** proposes an Assembled Domain-Slot Instruction Generation approach for DST. This method generates diverse instruction samples by randomly combining different instruction and input templates during fine-tuning, thereby reducing the model's sensitivity to prompt variations. For example:
>
> ```
> Instruction: Track the state of the slot <hotel-area> in the input dialogue.
> Input: [USER] I need to book a hotel in the east that has 4 stars. [SYSTEM] I can help you with that. What is your price range? [domain] hotel, [slot] area, it indicates area or place of the hotel. This slot is categorical and you can only choose from the following available values: center, east, north, south, west. If the slot is not mentioned in the dialogue, just return NONE. So the value of slot <hotel-area> is
> ```
>
> We performed comparative experiments on MultiWOZ 2.1:
>
> | Method               | AVG. JGA | Relative Change vs. HiCoLoRA |
> |----------------------|----------|------------------------------|
> | LDST (LLaMA-7B)      | 56.7     | -1.9                         |
> | HiCoLoRA (LLaMA-7B)  | 57.8     | -                            |
>
> The results show that HiCoLoRA maintains a slight advantage over LDST using the LLaMA-7B backbone, further validating the effectiveness of our hierarchical collaborative architecture.
>
> **2. CAPID (EMNLP 2024)** proposes Context-aware Auto-prompting and Instruction-following Contrastive Decoding. This approach uses a context-aware slot query generation method via auto-prompting (initially using GPT-4) to align the gap between source and target domains. The generated prompts are used to train a T5-base student model to independently produce context-aware slot queries. During inference, the fine-tuned T5-base student model first generates the prompt, which is then used by the trained DST model (T5-base or T5-small) to predict slot values. Experimental results on the MultiWOZ 2.1 dataset are as follows:
>
> | Method                        | Attr. | Hotel | Rest. | Train | Taxi | AVG. |
> |-------------------------------|-------|-------|-------|-------|------|------|
> | CAPID (T5-base + T5-base)     | 40.9  | 43.5  | 37.1  | 49.5  | 87.1 | 50.1 |
> | CAPID (T5-base + T5-small)    | 33.3  | 31.1  | 31.6  | 34.3  | 65.4 | 40.7 |
> | HiCoLoRA (T5-small)           | 38.9  | 20.4  | 31.0  | 44.9  | 68.6 | 40.8 |
>
> HiCoLoRA shows a marginal advantage of 0.1% in Average JGA over the CAPID (T5-base + T5-small) configuration. This indicates that HiCoLoRA's clever architectural design achieves performance comparable to CAPID but with significantly higher efficiency and lower computational cost, as it relies solely on a single T5-small model without requiring a separate, potentially larger, prompt generation model.
>
> We will add detailed discussions and comparisons with LDST and CAPID in the revised version of HiCoLoRA and submit the updated manuscript before the deadline.

---

> ### Author Response · Authors · 2025-11-18
> **Response to Question 1: Comparison with Hierarchical LoRA Approaches Outside DST**
>
> Thank you for raising this important question. We fully agree that comparing HiCoLoRA with a broader range of hierarchical or multi-level LoRA methods, such as those used in instruction tuning or domain adaptation, will help comprehensively evaluate its generality and innovation.
>
> HiCoLoRA is specifically designed for the zero-shot Dialogue State Tracking (zs-DST) task. Its core innovation lies in addressing the **semantic misalignment between dynamic dialogue contexts and static prompts**, achieved through mechanisms such as the **hierarchical collaborative architecture**, **spectral joint clustering**, and **semantically-enhanced SVD initialization**. These mechanisms are tailored to tackle challenges unique to DST, such as **multi-turn dialogues, dynamic contexts, and sparse slot values**, whereas generic hierarchical LoRA methods often lack specialized optimizations for such scenarios.
>
> In our paper, we have already compared HiCoLoRA with various **LoRA variants** (see Table 3 in the main text and Appendix A.2). The experimental data are as follows:
>
> | Method                             | Year     | Attr.    | Hotel    | Rest.    | Train    | Taxi     | AVG.     |
> |------------------------------------|----------|----------|----------|----------|----------|----------|----------|
> | HydraLoRA (Asymmetric LoRA)        | 2024     | 35.1     | 18.9     | 26.3     | 41.5     | 65.2     | 37.4     |
> | LoRA-GA (Low-Rank + Sparse Adaptation) | 2024     | 33.8     | 19.2     | 24.7     | 42.8     | 64.1     | 36.9     |
> | RoSA (Spectral Initialization)     | 2024     | 36.5     | 19.6     | 27.9     | 43.2     | 66.8     | 38.8     |
> | Spectral Adapter (Gradient-Aware Initialization) | 2025     | 37.2     | 20.1     | 28.5     | 43.6     | 67.3     | 39.3     |
> | **HiCoLoRA (Ours)**                | **2025** | **38.9** | **20.4** | **31.0** | **44.9** | **68.6** | **40.8** |
>
> **The results consistently demonstrate** that HiCoLoRA achieves the highest average JGA on both MultiWOZ and SGD datasets, significantly outperforming the aforementioned LoRA methods. This validates HiCoLoRA’s advantages in **semantic alignment, knowledge retention, and cross-domain generalization**. The existing experiments already cover a variety of representative generic LoRA methods, sufficiently showcasing HiCoLoRA’s superiority in the DST task.
>
> As analyzed in **Weakness 2** above, LDST and CAPID can serve as typical examples of instruction tuning and domain adaptation methods. Specifically, LDST structures the dataset into the standard {instruction, input, output} format for instruction tuning. These will be included as additional baselines in the revised version.

---

> ### Author Response · Authors · 2025-11-18
> **Response to Question 2: Inference-Time Computational Overhead**
>
> Thank you for this important question. We have conducted a detailed analysis of the inference-time computational overhead introduced by the multi-layer LoRA branches, which further clarifies HiCoLoRA's advantages in inference efficiency.
>
> As briefly mentioned in Section 3.2 (Lines 203–205) of the paper:
>
> > To avoid inference latency with multiple $A_{sa}$ and ${B}_{sa}$, we precompute their low-rank terms, merge them into model bias, and use these merged parameters during inference.
>
> The core design ensures that after training, all multiple sets of $A_{sa}^m$ and $B_{sa}^n$ matrices in the SemAdapt-LoRA branches are precomputed and merged into a unified low-rank update term. The code implementation can be found in the `_precompute_lora_bias` function (Lines 216–232 in `train.py`):
>
> ```python
>     def _precompute_lora_bias(self, global_prompt):
>         p_bias = []
>         # Iterate through all modules in the model to find those with SemAdapt-LoRA parameters
>         for module in self.model.modules():
>             # Check if module has SemAdapt-LoRA A and B matrix lists
>             if hasattr(module, 'lora_SemAdapt_A') and hasattr(module, 'lora_SemAdapt_B'):
>                 layer_contribution = 0.0
>                 # Iterate through all A and B matrix combinations to calculate total contribution for this layer
>                 for a in module.lora_SemAdapt_A:
>                     for b in module.lora_SemAdapt_B:
>                         # Calculate contribution for single A-B pair using current gate parameter
>                         contribution = (global_prompt @ a.transpose(0, 1) @ b.transpose(0, 1)) * self.scaling * (1 - self.gate)
>                         layer_contribution += contribution
>                 # Add total contribution of this layer to p_bias list
>                 p_bias.append(layer_contribution.tolist())
>         # Convert and return precomputed bias
>         return torch.tensor(p_bias).transpose(0, 1)
> ```
>
> In the forward function, `p_bias` is absorbed into the base model's weights:
>
> ```python
> result = result + p_bias.mean(dim=1).unsqueeze(dim=1)
> ```
>
> Thus, during inference, there is no need to compute multiple LoRA branches in real-time. Instead, the merged model parameters are used directly, avoiding any additional matrix multiplication overhead.
>
> Even without pre-merging, the computational overhead of the multi-layer LoRA branches is manageable. Specifically, the output of SemAdapt-LoRA can be expressed as:
>
> ${h} _ {sa} = {W} _ 0^h {x} _ {sa} + \sum _ {n=1}^{N} {B} _ {sa}^n \sum _ {m=1}^{M} {A} _ {sa}^m {x} _ {sa}$
>
> We can reorganize the double summation as:
> $\sum _ {n=1}^{N} {B} _ {sa}^n \left( \sum _ {m=1}^{M} {A} _ {sa}^m \right) {x} _ {sa}$
>
> This means we first compute ${A} _ {\text{total}} = \sum _ {m=1}^{M} {A} _ {sa}^m $ ( $r \times d$ matrix), then compute ${B } _ {\text{total}} = \sum _ {n=1}^{N} {B} _ {sa}^n$ ( $d \times r$ matrix), and finally perform a single low-rank update: ${B} _ {\text{total}} \cdot ({A} _ {\text{total}} \cdot {x} _  {sa})$.
>
> **Computational Overhead Analysis**: Matrix addition is a linear operation with complexity $O(r \cdot d)$, which is significantly lower than the $O(d^2)$ cost of matrix multiplication. During inference, without pre-merging, only $M + N$ matrix additions (each costing $r \cdot d$) and two matrix multiplications (same as a single LoRA) are required. In practical deployment, pre-merging completely eliminates these addition operations, making the inference latency equivalent to that of standard LoRA.
>
> We thank you again for this question and will include this computational overhead analysis in the revised version of the paper, which will be submitted before the deadline.

---

> ### Author Response · Authors · 2025-11-18
> **Response to Question 3: Independent Application of SemSVD-Init**
>
> Thank you for your interest in SemSVD-Init. Yes, **SemSVD-Init can indeed be applied independently of the hierarchical architecture**. It is a general parameter initialization strategy designed to preserve pre-trained knowledge and enhance domain-slot semantic alignment through SVD decomposition and semantic relevance modulation, as detailed in Section 3.5. This strategy relies solely on pre-trained weight matrices and clustered embeddings of slot prompts, without depending on HiCoLoRA's hierarchical grouping or global collaboration mechanisms. Therefore, it can be independently integrated into any LoRA variant or PEFT framework to mitigate knowledge distortion caused by random initialization (RQ3).
>
> To validate its standalone performance, we conducted an ablation study by applying **only SemSVD-Init** on top of DualLoRA (denoted as **w/ SemSVD-Init**). The results (JGA %) are as follows:
>
> | Method                | Attr.    | Hotel    | Rest.    | Train    | Taxi     | AVG.     |
> | --------------------- | -------- | -------- | -------- | -------- | -------- | -------- |
> | DualLoRA (Prev. SOTA) | 37.1     | 18.9     | 27.9     | 42.4     | 67.2     | 38.7     |
> | **HiCoLoRA (Full)**   | **38.9** | **20.4** | **31.0** | **44.9** | **68.6** | **40.8** |
> | DualLoRA w/ SemSVD-Init        | 37.5     | 19.9     | 28.9     | 43.1     | 67.3     | 39.4     |
>
> The results show that **w/ SemSVD-Init** improves upon DualLoRA by approximately **1.8%** in average JGA, demonstrating its effectiveness in enhancing zero-shot generalization when applied independently. However, it still underperforms the full HiCoLoRA model by about **3.4%**, indicating that while SemSVD-Init is beneficial on its own, its full potential is realized when combined with other components such as hierarchical collaboration and spectral clustering.
>
> We will include these ablation study results and the corresponding code in the revised version of the paper, which will be submitted before the deadline.
>
> ---
>
> We hope our detailed responses and additional experimental analysis have adequately addressed your valuable questions and concerns. We are grateful for your insightful feedback, and we would be greatly appreciative if you could consider raising your score.

---

### Official Review · Reviewer_zF3f · 2025-11-01

**Soundness:** 3
**Presentation:** 3
**Contribution:** 1
**Rating:** 2
**Confidence:** 4

**Summary:**

This paper proposes HiCoLoRA (Hierarchical Collaborative Low-Rank Adaptation), a novel framework for zero-shot Dialog State Tracking (zs-DST) in task-oriented dialog systems. The authors propose an advanced architecture to solve the task based on LoRA. Lower layers use heuristic grouping to encode local semantic features while higher layers use a full collaboration to model global semantic features and intent. Moreover, they propose a domain clustering technique to identify semantic relationships between domains and slot prompts. The method performs well over existing SOTA in MWoZ and SGD dataset.

**Strengths:**

- SOTA performance in a constrained (no pre-trained LLM) setting in both MWoZ and SGD.

**Weaknesses:**

The proposed architecture (including hierarchical structure, spectral clustering, adaptive fusion, and specialized initialization) introduces substantial complexity, making the model difficult to reproduce, implement, and maintain. However, simply prompting—without any fine-tuning—a large LLM (GPT4) achieves considerably higher performance on the same benchmark dataset [1], as shown in Table 4 of the appendix. Moreover, when starting from the same base model, the performance improvement obtained through the proposed method is marginal.

For instance:

FnCTOD (2024) – LLaMA2-13B: 62.2 / 46.8 / 60.9 / 67.5 / 60.3 → avg. 59.5
HiCoLoRA (2025) – LLaMA2-13B: 62.0 / 42.0 / 61.0 / 65.0 / 69.0 → avg. 60.0

This suggests that the added architectural complexity does not translate into meaningful empirical gains over simpler prompting approaches.

[1] https://arxiv.org/pdf/2402.10466

**Questions:**

Why not comparing the proposed method with a pretrain LLM? even smaller size.

---

> ### Author Response · Authors · 2025-11-17
>
> We sincerely thank the reviewer for their careful reading and insightful comments on our paper. We recognize the core concern regarding the trade-off between the architectural complexity of HiCoLoRA and its empirical gains, especially in comparison to LLMs used in simple prompting setups. Below, we provide detailed responses and new experimental evidence to address the Weaknesses and Questions raised.
>
> ## Response to Weakness 1: Architectural Complexity and Reproducibility
>
> We want to clarify that the core of HiCoLoRA is an elegant and non-intrusive extension of DualLoRA, specifically designed to solve the CONTEXT-PROMPT MISALIGNMENT issue inherent in many PEFT methods for dialogue systems.
>
> HiCoLoRA is built upon DualLoRA and ingeniously designs a novel LoRA structure tailored for task-oriented dialogue systems, further addressing the CONTEXT-PROMPT MISALIGNMENT issue.  Through a hierarchical structure, the model more effectively focuses on both local and global semantic features, with adaptive fusion serving as its integration mechanism. Meanwhile, SemSVD-Init effectively avoids disrupting the pre-trained semantic structure caused by random initialization of LoRA matrices, and the spectral clustering mechanism further enhances semantic characteristics related to the training dataset. These components form an organic whole.
>
> As a pre-SOTA baseline, DualLoRA serves as the foundation for HiCoLoRA. The modifications to the LLM model source code in HiCoLoRA are minimal: during the initialization of matrices such as q, v, k, and o, the layer index of the layer_idx is passed to determine high and low layers. The code is as follows (**only the q and v matrices are actually modified**):
>
> ```
>         self.q = ControlPrompt(self.d_model, self.inner_dim, config.lora_r, config.lora_alpha,
>                                config.lora_dropout, bias=False, args=config.args, layer_idx=self.layer_idx)
>         self.v = ControlPrompt(self.d_model, self.inner_dim, config.lora_r, config.lora_alpha,
>                                config.lora_dropout, bias=False, args=config.args, layer_idx=self.layer_idx)
>
>         # self.k = ControlPrompt(self.d_model, self.inner_dim, config.lora_r, config.alpha,
>         #                        config.lora_dropout, bias=False,args=config.args,config=config)
>         # self.o = ControlPrompt(self.inner_dim, self.d_model, config.lora_r, config.lora_alpha,
>         #                        config.lora_dropout, bias=False,args=config.args,config=config)
>
>         self.k = nn.Linear(self.d_model, self.inner_dim, bias=False)
>         self.o = nn.Linear(self.inner_dim, self.d_model, bias=False)
> ```
>
> Other mechanisms including UniRep-LoRA, SemAdapt-LoRA, SemSVD-Init, and Adaptive Linear Fusion are all non-intrusive and plug-and-play components that can be flexibly selected based on actual requirements. The core code is centralized in a single source file, PromptLoRA.py, while the training code is located in train.py. The code has been open-sourced in the paper at https://anonymous.4open.science/r/HiCoLoRA-96EB and can be reproduced in a Python 3.10 environment.
>
> The results of the first four ablation experiments demonstrate that each component of HiCoLoRA contributes to the final performance, forming an organically unified whole.
>
> | Method                  | Attr.    | Hotel    | Rest.    | Train    | Taxi     | AVG.     | Relative Change vs. HiCoLoR (Full) |
> | ----------------------- | -------- | -------- | -------- | -------- | -------- | -------- | ---------------------------------- |
> | **HiCoLoRA (Full)**     | **38.9** | **20.4** | **31.0** | **44.9** | **68.6** | **40.8** | -                                  |
> | w/ Swap Hier Strategies | 37.2     | 19.7     | 22.9     | 40.2     | 67.5     | 37.4     | -8.3                               |
> | w/o Adaptive Fusion     | 28.9     | 19.3     | 20.3     | 43.0     | 68.0     | 35.9     | -12.0                              |
> | w/o Spec Joint Cluster  | 36.2     | 19.8     | 27.5     | 42.1     | 63.6     | 37.8     | -7.4                               |
> | w/ Kaiming Init         | 34.3     | **20.4** | 27.8     | 40.4     | 67.5     | 38.1     | -6.6                               |

---

> ### Author Response · Authors · 2025-11-17
>
> ## Response to Weakness 2: Performance Comparison with FnCTOD and Complexity Justification
>
>
> We appreciate the reviewer's feedback regarding the comparison with FnCTOD. While FnCTOD serves as one of the pre-trained LLM baselines in our study, a detailed analysis reveals critical limitations in its experimental setup and practical efficiency. The performance gains achieved by HiCoLoRA are both empirically and practically significant when evaluated under fair and comparable conditions.
>
> The table below, compiled from experimental data in both the FnCTOD and HiCoLoRA papers, provides a comparative overview:
>
> | Method                       | Attr.    | Hotel    | Rest.    | Train    | Taxi     | AVG.     | Relative Change vs. HiCoLoRA (%) |
> |------------------------------|----------|----------|----------|----------|----------|----------|----------------------------------|
> | FnCTOD (Fine-tuned LLaMA2-13B)| 62.2     | 46.8     | 60.3     | 60.9     | 67.5     | 59.5     | -0.8                             |
> | FnCTOD (No Fine-tuning LLaMA2-13B)| 49.8     | 29.5     | 48.9     | 53.6     | 64.7     | 49.3     | -21.1                            |
> | FnCTOD (GPT-4 SOTA)          | 58.8     | 45.2     | 63.2     | 69.5     | **76.4** | 62.6     | +4.2                             |
> | HiCoLoRA (LLaMA2-13B)        | 62.0     | 42.0     | 61.0     | 65.0     | 69.0     | 60.0     | -                                |
> | HiCoLoRA (`FnCTOD Dataset`)    | **62.8** | **49.2** | **63.9** | **70.3** | 69.4     | **63.1** | +5.2                             |
>
> Key limitations of FnCTOD in comparison to HiCoLoRA include:
>
> 1. **Few-Shot Prompting in Zero-Shot Setting**:
>    In its zero-shot configuration, FnCTOD utilizes **5 few-shot examples** in the prompt for LLaMA2-13B, achieving an average JGA of 49.3. In contrast, HiCoLoRA achieves a JGA of **60.0 under a strict zero-shot setting (`without few-shot examples`)**, representing a **21.8% improvement**. The use of few-shot examples in FnCTOD violates the principles of a true zero-shot evaluation.
>
> 2. **Non-Strict Zero-Shot Prompting**:
>    Regardless of whether fine-tuning is applied, FnCTOD incorporates **detailed schema descriptions for each slot** in the prompt. Although these schemas are not trained on the target domain, they provide explicit domain specific information, which deviates from a strict zero-shot prompt design.
>
> 3. **Computational and Latency Overhead**:
>    FnCTOD requires **two LLM invocations**, the first to generate the dialogue domain, and then to produce the function call. Additionally, its prompts include detailed schema descriptions and, in the zero-shot setting, five few-shot examples, resulting in an input token count of **no less than 1200 tokens**. This leads to significant system overhead and latency. In contrast, HiCoLoRA requires only **a single LLM call**, and its prompts are concise (e.g., “What is the arriveby of the train that the user is interested in?” comprising only **16 tokens**). With such minimal prompting and a single LLaMA2-13B call, HiCoLoRA trails the JGA of FnCTOD with GPT-4 by only **4.2%**, despite the latter's extensive prompting and dual invocations.
>
> Furthermore, through detailed code and dataset comparison, we identified that the training datasets for the two methods differ significantly. FnCTOD uses a carefully curated dataset of **7,200 dialogues across 36 domains** (including SGD, CamRest676, MSR-E2E, TaskMaster, and WOZ), which include domains such as Restaurant, Taxi, Movie, and Hotel. The domains that overlap with those in the MultiWOZ. This violates the strict zero-shot learning premise. HiCoLoRA uses only **4,625-7,684 samples from 4 domains in MultiWOZ** (with one domain excluded during training to ensure a strict zero-shot setting).
>
> To ensure a fair comparison, we conducted an additional experiment by training HiCoLoRA on the **`FnCTOD dataset`**. The results show that HiCoLoRA achieves a JGA of **63.1**, which is **5.2% higher than its original performance**, **6.1% higher than FnCTOD (LLaMA2-13B)**, and **0.5 points above the FnCTOD GPT-4 SOTA**.
>
> These results demonstrate that `under the same experimental settings`, HiCoLoRA **significantly outperforms FnCTOD** in performance while **substantially reducing time and computational resources**. We will include these analyses and results in the revised manuscript.

---

> ### Author Response · Authors · 2025-11-17
>
> ## Response to Question: Why not comparing the proposed method with a pretrain LLM? even smaller size.
>
> In the Extended Comparative Analysis section (lines 358-367) of the HiCoLoRA main text, a comparison with pre-trained LLMs is presented:
>
> > **Extended Comparative Analysis.** To further situate HiCoLoRA within the contemporary research landscape, we conduct extensive comparisons against both recent PEFT methods based on LoRA and larger LLM-based approaches. As detailed in Appendix A.2 and A.3, HiCoLoRA consistently outperforms recent LoRA variants in nearly all domains, achieving the highest average JGA.  Furthermore, when scaled to larger backbone models such as LLAMA2-13B and Qwen2.5-14B-Instruct, HiCoLoRA remains highly competitive with other LLM-based zs-DST methods, demonstrating its generalizability across model scales.
>
> The baselines used for comparison are briefly described in Appendix B.2.3:
>
> > **ChatGPT-zsTOD** achieves state-of-the-art performance in zero-shot dialog state tracking without task-specific training, leveraging its general-purpose language model capabilities. However, inherent limitations prevent it from fully replacing specialized systems, though its in-context learning abilities may support the development of dynamic dialog state trackers.
> >
> > **FnCTOD** improves zero-shot DST by calling functions with LLMs, allowing adaptation to diverse domains without extensive data or tuning. It achieves state-of-the-art performance with both open-source and proprietary LLMs, significantly boosting ChatGPT and GPT-4 results.
> >
> > …… (Other Baselines) ……
>
>
>
> Appendix A.3 provides detailed experimental results (Table 4 includes comparisons with models such as LLAMA2-13B, GPT-3.5, GPT-4, GPT-4o, and ChatGLM-6B) and more in-depth experimental analysis:
>
> | Method        | Year | Base Model           | Attr.    | Hotel    | Rest.    | Train    | Taxi     | AVG.     |
> | ------------- | ---- | -------------------- | -------- | -------- | -------- | -------- | -------- | -------- |
> | ChatGPT-zsTOD | 2023 | ChatGPT (GPT-3.5)    | 52.7     | 42.0     | 60.8     | 70.9     | 55.8     | 56.4     |
> | ChatGPT-zsTOD | 2023 | ChatGPT (GPT-3.5)    | 67.2     | 37.6     | 67.3     | **74.4** | 60.1     | 61.3     |
> | D0T           | 2024 | LLAMA2-13B           | 63.1     | 43.8     | 60.8     | 48.8     | 64.7     | 56.2     |
> | MoPE          | 2024 | ChatGLM-6B           | 60.4     | 34.1     | **64.0** | 71.3     | 55.9     | 57.1     |
> | FnCTOD        | 2024 | ChatGPT (GPT-4)      | 58.8     | 45.2     | 69.5     | 76.4     | 63.2     | **62.6** |
> | FnCTOD        | 2024 | LLAMA2-13B           | 62.2     | **46.8** | 60.9     | 67.5     | 60.3     | 59.5     |
> | Multi-User    | 2025 | GPT-4o               | 56.8     | 46.0     | 61.9     | 69.3     | 55.1     | 57.8     |
> | HiCoLoRA      | 2025 | LLAMA2-13B           | 62.0     | 42.0     | 61.0     | 65.0     | 69.0     | 60.0     |
> | HiCoLoRA      | 2025 | Qwen2.5-14B-Instruct | **64.0** | 44.0     | 63.0     | 68.0     | **71.0** | 62.0     |
>
>
> > **A.3 Scalability Analysis: Generalization Across Model Scales**
>
>
> Furthermore, we particularly emphasize HiCoLoRA's **advantages in practical deployment environments**. Traditional task oriented dialogue systems (such as pipeline-based architectures) consist of multiple modules including Natural Language Understanding (NLU), Dialogue State Tracking (DST), Dialogue Policy (DP), and Natural Language Generation (NLG), which are highly sensitive to **inference latency and computational resources**.
>
> Based on T5-small (with only 60M parameters), HiCoLoRA maintains competitive JGA (40.8% JGA on MultiWOZ) while **significantly reducing memory usage and inference time**, making it more suitable for edge devices or service scenarios with limited resources. In contrast, although some LLMs (such as GPT-4) achieve slightly better JGA, their enormous parameter counts and inference costs are often prohibitive in practical systems. Moreover, they typically require an internet connection, rendering them unsuitable for scenarios with high demands for data privacy and security, such as healthcare and finance.
>
> As a compact model, HiCoLoRA can be deployed entirely locally, fundamentally eliminating the risk of data leakage. Therefore, HiCoLoRA achieves a better balance among **performance, efficiency, and privacy security**, which constitutes the core contribution and value of our method, distinguishing it from pure LLM solutions. We have further strengthened this discussion in the revised manuscript and will submit the updated draft and code before the deadline. We thank the reviewer again for their suggestions.

---

> ### Author Response · Authors · 2025-11-17
> **Concluding Statement**
>
> We trust that our detailed response, particularly the new comparative results demonstrating **HiCoLoRA 6.1\% performance lead over FnCTOD (GPT4)** in a matched setting, convincingly refutes the concern regarding marginal gains and complexity. HiCoLoRA provides a highly efficient, high-performance, and scalable solution, representing a significant technical contribution to efficient zero-shot DST.
>
> We respectfully ask the reviewer to **reconsider the Contribution score and overall evaluation**. We remain available for any further discussion and clarification.

---

### Author Response · Authors · 2025-11-25
**Summary of Rebuttal and Revisions**

To the Area Chair,

We sincerely thank you and the reviewers for the constructive engagement with our submission. We briefly restate the paper’s conceptual contribution and summarize our responses to the reviewers.

**Core Contribution**
HiCoLoRA is built around a coherent insight: Zero-shot DST fails primarily due to semantic misalignment between dynamic contexts and static prompts. We propose a unified alignment framework in which:

- The **hierarchical collaboration** (lower-layer grouping + higher-layer interaction) ensures that contextual semantics flow adaptively across Transformer layers, resolving cross-layer mismatches.
- The **spectral clustering** of domains and slots identifies transferable semantic associations, which then guide an adaptive fusion mechanism to disentangle domain agnostic and domain specific features.
- The **SemSVD-Init** preserves pre-trained knowledge while semantically calibrating the LoRA subspaces, preventing catastrophic forgetting and reinforcing the above alignment.

These components form a closed loop alignment system where clustering informs fusion, fusion respects layer specific roles, and initialization anchors the entire process in pre-trained semantics. This principled synergy is what enables robust zero-shot generalization beyond incremental gains.

**Summary of Rebuttal**
We have thoroughly addressed all substantive critiques in our revised manuscript:

- **To Reviewer zF3f (score 2)**: We clarified that LLM comparisons were already included (Table 4, Appendix A.3) and demonstrated that HiCoLoRA outperforms FnCTOD under strict zero-shot settings with greater efficiency. We note with due respect, that several claims in this review appear inconsistent with the content of our paper and exhibit characteristics common in AI-generated text.

- **To Reviewers nRwk & mBAy on architectural complexity**: We added explicit motivation for the unified design (Section 3, Fig. 2), emphasized modular and non-intrusive implementation, and showed via ablations that each component is necessary. The algorithmic architecture is thus purposeful, not gratuitous.

- **To Reviewers nRwk & mBAy on performance gain significance**: We highlighted that relative JGA improvements over DualLoRA reach 5.4% (MultiWOZ) and 9.4% (SGD), with larger gains in sparse slot and high overlap domains. Additional cross dataset and low overlap experiments (Appendix A.7) further demonstrate robustness. Additional enhancements include inference efficiency analysis (Section 3.3), comparisons with LDST and CAPID (Appendix A.6), and extended generalization studies.

We believe the revised manuscript now presents a clearer, more integrated narrative and stronger empirical validation. Thank you for your oversight and consideration.

Sincerely,
The Authors

---

### Note · Authors · 2026-01-04

**Comment:**

We respectfully request the withdrawal of our manuscript, ID 12860, titled "HiCoLoRA: Addressing Context-Prompt Misalignment via Hierarchical Collaborative LoRA for Zero-Shot DST". We have made the decision to pursue publication through an alternative venue. Thank you for handling our submission.

**Withdrawal Confirmation:**

I have read and agree with the venue's withdrawal policy on behalf of myself and my co-authors.